# Asian dust transport proteinaceous matter from the Gobi Desert to Northern China

Ren-Guo Zhu[1,2], Hua-Yun Xiao[3, *], Meiju Yin[3], Hao Xiao[3], Zhongkui Zhou[1], Yuanyuan Pan[1], Guo Wei[1], Cheng Liu[1]

[1]School of Water Resources and Environmental Engineering, East China University of Technology, Nanchang 330013, China.

[2]Jiangxi Provincial Key Laboratory of Genesis and Remediation of Groundwater Pollution, East China University of Technology, Nanchang 330013, China.

[3]School of Agriculture and Biology, Shanghai Jiao Tong University, Shanghai 200240, China.

*Corresponding author:* Hua-Yun Xiao (Xiaohuayun@sjtu.edu.cn)

**Abstract.** Asian dust can greatly influence the ecosystem's productivity and biogeochemical cycles by providing new nutrients. However, the transport of proteinaceous matter (combined amino acids, CAAs) by Asian dust to downwind ecosystems remains unclear. Here, the concentrations and $\delta^{15}N$ values of individual CAAs in Gobi surface soil and vegetation, as well as in $PM_{2.5}$ samples from four cities in Northern China were characterized. Proline dominated the total pool of CAAs in urban $PM_{2.5}$ during non-dust periods, whereas CAAs transported by Gobi dust were rich in alanine, glycine, and glutamic acid. The concentrations and percentages of these three CAAs in $PM_{2.5}$ from Northern China notably increased during dust periods. During non-dust periods, the $\delta^{15}N$ values of individual CAAs in urban $PM_{2.5}$ fell within their respective ranges in local urban sources, suggesting CAAs in $PM_{2.5}$ were primarily influenced by local urban sources during non-dust periods. Compared to their values in urban $PM_{2.5}$ during non-dust periods, glycine and leucine in Gobi Desert sources exhibited $\delta^{15}N$ depletion by more than 6‰. During dust periods, glycine and leucine in urban $PM_{2.5}$ all exhibited negative shifts in their $\delta^{15}N$ values, confirming that Gobi dust is a significant source of CAAs in $PM_{2.5}$ in Northern China. The dry deposition of protein-N from Gobi dust was calculated using nitrogen isotopic mass balance based on the $\delta^{15}N$ values of glycine and leucine, yielding a value of up to 0.36 mg N $m^{-2}$ $d^{-1}$. The rapid accumulation of such considerable protein-N quantities may profoundly affect oligotrophic ecosystem productivity.

**1 Introduction**

Asian dust, originating from the desert and Gobi regions of Inner Mongolia in China and southern

Mongolia, are a significant source of atmospheric particulate matter, annually emitting approximately

1000-3000 teragrams (Tegen and Schepanski, 2009). Recent studies indicate that, in addition to mineral

dust, these storms can transport substantial amounts of water-soluble organic nitrogen (WSON) to urban

aerosols and even distant oceans (Liu et al., 2021; Mochizuki et al., 2016; Tsagkaraki et al., 2021).

Importantly, primary biological particles present in desert soils can be lifted by dust particles into the

stratosphere and transported over long distances (Favet et al., 2013).

Combined amino acids (CAAs), including proteins and peptides, are important constituents of

atmospheric organic nitrogen (Jaenicke, 2005; Zhang and Anastasio, 2003). Recent studies reveal that

they are ubiquitous in aerosols across various environments such as urban, suburban, rural, and remote

areas (Li et al., 2022a, b). Protein-containing particles are expected to influence particle hygroscopicity,

atmospheric chemistry, cloud formation processes and new particle formation and growth (Chan et al.,

2005; Elbert et al., 2006; Haan et al., 2009; Jaber et al., 2021; Violaki and Mihalopoulos, 2010).

Moreover, they serve as bioavailable nitrogen sources and greatly contribute to the biogeochemical

nitrogen cycles (Matsumoto et al., 2017; Neff et al., 2002). Therefore, the nutrient enrichment resulting

from dry deposition of proteinaceous matter attract attention in aerosol studies (Samy et al., 2013; Xu et

al., 2020; Zhang and Anastasio, 2003).

However, the sources of CAAs in aerosols remain insufficiently elucidated. Generally, the suspension

process of biological particles, including fungi, spores, bacteria, molds, animal dander, pollen, and

fragments of plants and animals, along with soil particles, has been identified as primary sources of CAA

in aerosols (Filippo et al., 2014; Matos et al., 2016; Samy et al., 2013). Besides that, atmospheric proteins

are also suggested to originate from biomass burning (Kang et al., 2012; Song et al., 2017; Zhu et al.,

2020a) and marine sources, such as bursting sea bubbles and suspended algae (Feltracco et al., 2019; Li

et al., 2022a; Triesch et al., 2021). However, Matsumoto et al. (2021) found a significant correlation

between the total CAAs concentration in fine particles at an urban site in Japan and the concentration of

non-sea-salt calcium (nss-$Ca^{2+}$), suggesting an influence of Asian dust particles on airborne CAAs. To

date, no studies have directly compared the protein characteristics in $PM_{2.5}$ from urban environments

with those in dust sources to conclusively determine the contribution of dust sources to proteinaceous materials in downwind urban $PM_{2.5}$ during dust events.

Composition profiles of CAAs have been utilized to identify emission sources of primary biological aerosol particles (Abe et al., 2016; Matsumoto et al., 2021). Zhang and Anastasio (2003) noted a significant contribution of serine to the total CAAs pool in the $PM_{2.5}$, highlighting the direct emissions from biological sources such as plants and animals. Matsumoto et al. (2021) identified glutamic acid, glycine, and aspartic acid as the dominant amino acids in the total CAAs pool in fine aerosols, indicating that regional and locally derived biomass burning and fossil fuel combustion are the sources of CAAs in fine atmospheric particles. Moreover, our previous study suggested distinct differences between the CAA composition profiles in surface soil and plant sources; hydrophobic species (alanine, valine, leucine, and isoleucine) and neutral proline were the major CAA species in soil sources, while hydrophilic CAA species (glutamic acid, lysine, and aspartic acid) represented major fractions in plant sources (Zhu et al., 2020b). Unfortunately, the composition profiles of CAAs in distant dust sources and local urban aerosols remain inadequately characterized.

With the development of stable N isotope technology, compound-specific isotope analysis of amino acids has become an effective tool to trace the sources and cycling of dissolved organic nitrogen in marine environment (Batista et al., 2014; Ianiri and McCarthy, 2023; Yamaguchi and McCarthy, 2018). In atmospheric studies, $\delta^{15}N$ values of glycine have been employed as a novel method to identify the sources of proteinaceous matter in aerosols (Zhu et al., 2021). Our previous study suggested that the $\delta^{15}N$ values of Gly (glycine) in aerosol particles from biomass burning sources (averaged +22.4±4.4‰) were more positive than those of soil (averaged +5.2±3.5‰) and plant sources (averaged -13.4‰) (Zhu et al., 2020b). Therefore, according to the $\delta^{15}N$ inventories of specific CAAs in potential emission sources, the main sources of CAAs in atmospheric particles could be identified. Changes in $\delta^{15}N$ values of specific CAAs in the aerosols could be close related to the variation in the primary emission sources of atmospheric proteinaceous matter (Zhu et al., 2020b). However, limited knowledge on the $\delta^{15}N$ inventories of individual CAAs in dust sources and local urban sources complicates the elucidation of CAA origins in urban $PM_{2.5}$ on dusty days.

East Asia is the second-largest dust source region worldwide, with annual dust emissions estimated at 214 Tg yr$^{-1}$ (Tian et al., 2020). Furthermore, the westerlies in the Northern Hemisphere's middle latitudes

can transport Asian dust from upwind areas to distant downwind regions, potentially completing a global cycle and exerting far-reaching impacts (Xie et al., 2023; Zhou et al., 2019). Asian dust particles primarily originate from the arid and semi-arid areas of northwestern China and Mongolia, including the Taklamakan Desert, the Gobi Desert, the Badan Jaran Desert, the Tengger Desert, and others (Shao and Dong, 2006). Recent research indicates that the Gobi Desert, rather than the Taklamakan Desert, is the primary contributor to dust concentrations in East Asia in spring (Chen et al., 2017; Tang et al., 2018). Spring is considered the peak season for sand and Asian dust events in Northeast Asia, as positive surface pressure anomalies over the Tamil Peninsula intensify cold air outbreaks across the desert regions of northwestern China and Mongolia (Yang et al., 2008). A particularly intense and widespread dust event occurred between March 26–29, 2018, in the North China Plain, which was the most significant Asian dust event in recent years in China (Zhou et al., 2019). This dust event affected nearly two-thirds of China and parts of the Northwest Pacific (Tian et al., 2020). Therefore, $PM_{2.5}$ sampling at four representative sites (Beijing, Tianjin, Shijiazhuang, and Taiyuan) located in the downwind areas of the Gobi Desert during the 26–29 March 2018 dust event provides a typical representation of Asian dust activity in northern China.

The overall goal of this present study was to evaluate the contribution of the Gobi dust sources to the proteinaceous matter (CAAs) in $PM_{2.5}$ from four representative urban centers in Northern China on dusty days. Firstly, we analyzed the concentrations and $\delta^{15}N$ values of individual CAAs in the surface soil and prevalent plants in the Gobi Desert, along with surface soil and the predominant plant species in Beijing, Tianjin, Shijiazhuang, and Taiyuan. The objective was to examine the composition characteristics and nitrogen isotope signature of individual CAAs from both Gobi dust sources and local urban sources; Secondly, variations in the concentrations and $\delta^{15}N$ values of CAAs in $PM_{2.5}$ samples from these four cities during non-dust and dust periods were analyzed to confirm whether Gobi dust sources contribute to CAAs in $PM_{2.5}$ in Northern China and to quantify the extent of this contribution for each city. Finally, we quantified the dry deposition of protein-N transported by the Asian dust. This work could potentially improve current knowledge on the influence of Asian dust on productivity and biogeochemical nitrogen cycles in downwind ecosystems.

## 2 Materials and Methods

### 2.1 Sample collection

An intensive ground monitoring network consisting of four sites (Beijing, Tianjin, Shijiazhuang, and Taiyuan) in Northern China was set up to monitor the dust episodes (Figure S1). The MODIS satellite image (https://worldview.earthdata.nasa.gov/), as shown in Figure S2, shows a dust episode with brown dust plumes clearly visible over these four sampling sites.

The $PM_{2.5}$ samples on quartz fiber filters were simultaneously collected at Beijing, Tianjin, Shijiazhuang, and Taiyuan by a high-volume air sampler (KC-1000, Qingdao Laoshan Electronic Instrument company, China) at a flow rate of $1.05 \pm 0.03$ $m^3$ $min^{-1}$. The high-volume air sampler was set on the rooftop of a building (approximately 12 m above ground) at each site. Quartz filters were pre-combusted at 450 °C for 10 hours; then wrapped in pre-combusted (450 °C, 10hr) aluminum foil envelopes and placed in separate plastic bags. Daily $PM_{2.5}$ samples were collected from March 24, 2018 to March 31, 2018 at four cities. The sampling duration time for each sample was generally 23.5 h from 9:00 to 8:30 LT (local time) of the next day. All sample filters were sealed individually in an aluminum foil bag and stored at -20°C prior to analysis. Field blank samples were also collected and analyzed as the control.

Surface soil samples (n=5) and leaves from three common plants (n=12), *chenopodium ambrosioides, Tripogon chinensis, and tamarix chinensis*, were collected in the Gobi area (Table S1), ranging between 43.46°N to 43.60°N and 112.00°E to 112.05°E, covering approximately 45 km², as shown in Figure S1. The Gobi Desert is the major source of sand for Asian dust events in Asia during the spring (An et al., 2013). Therefore, five sampling sites along the transport pathway of the dust event that occurred from March 26 to 29, 2018 (Figure 2), were selected to represent the Gobi Desert. Each site was free of anthropogenic interference. Surface sand samples, which are most likely to be aerosolized, were collected from the tops of dunes using a plastic spatula and stored in sealed plastic bags until transported to the laboratory. At each location, surface soil was collected from five randomly selected sampling points within a radius of approximately 20 cm. These five sub-samples were then combined to create one representative sample.   Additionally, to examine the local sources of combined amino acids (CAAs) in $PM_{2.5}$, surface soil samples    and leaves from the predominant plant, *Platanus orientalis*, were collected in Beijing, Tianjin, Shijiazhuang, and Taiyuan.

## 2.2 Analyses of the concentration and δ$^{15}$N value of CAAs

Extraction methods for combined amino acids (CAAs) in PM$_{2.5}$ were detailed in our previous study (Zhu et al., 2020b). To convert all amino acids to free amino acids, a hydrolysis method was employed. The CAA concentrations were determined by subtracting free amino acid concentrations from total amino acid concentrations. Briefly, one-sixteenth of each filter sample (~80 m$^3$ of air) was cut into small pieces and hydrolyzed using 10 mL of 6 M hydrochloric acid at 110°C for 24 h. α-Aminobutyric acid was added as an internal reference. To prevent oxidation, 25 μL of 20 μg μL$^{-1}$ ascorbic acid (500 μg absolute) was added to each filter sample, and each hydrolysis tube was flushed with nitrogen and tightly sealed before hydrolysis. The hydrolyzed solution was cooled, dried under a stream of nitrogen, and then redissolved in 0.1 N HCl (v/v). The extracts of total amino acids were then purified using a cation exchange column (Dowex 50W X 8H$^+$, 200-400 mesh; Sigma–Aldrich, St Louis, MO, USA), eluted with 10 mL of 10% aqueous ammonia, and dried under nitrogen. Finally, tert-butyldimethylsilyl derivatives of total amino acids were prepared as described in Zhu et al. (2018).

Free amino acids were extracted following the procedure outlined in Zhu et al. (2020b), where one-quarter of each filter sample (~300 m$^3$ of air) was processed in a Nalgene tube with α-aminobutyric acid as the internal standard and ultrasonically extracted in ice-cold Milli-Q water. The extract underwent ultrasonication, shaking, centrifugation, and filtration through a 0.22 μm cellulose acetate membrane, followed by lyophilization and resuspension in 1 mL of 0.1 N HCl (v/v). The samples were then processed using the same purification and derivatization steps as the total amino acids. Field blank filters were also taken and treated using the same procedure, and all reported values were corrected for blanks.

Plant and soil samples were ground separately in liquid nitrogen into fine powders using a mortar and pestle. Then, well ground and homogenized soil and plant power were hydrolyzed, purification and derivatization in the same way as the aerosol samples. For more details refer to our previous publication (Zhu et al., 2021).

The concentrations of CAAs were analyzed using a gas chromatograph–mass spectrometer (GC–MS). The GC–MS instrument was composed of a Thermo Scientific TRACE GC (Thermo Scientific, Bremen, Germany) connected to a Thermo Scientific ISQ QD single quadrupole MS. More details on instrument conditions, quality assurance and control (including recoveries, linearity, detection limits and quantitation limits) of CAAs are provided in our previous publications (Zhu et al., 2018, 2020b). To

evaluate the extraction efficiency, analytical method was applied to the samples were spiked with the amino acid standard mixtures (100μl 1nmol μl$^{-1}$). The average recovery ratios of the individual amino acids were shown in Table S2. The recoveries for the majority of CAAs ranged from 80.7% (tyrosine) to 106.5% (glycine). The precisions of the investigated AAs were better than 10%.

The δ$^{15}$N values of AA-tert-butyldimethylsilyl derivatives were analyzed using a Thermo Trace GC (Thermo Scientific, Bremen, Germany) and a conflo IV interface (Thermo Scientific, Bremen, Germany) interfaced with an isotope ratio mass spectrometry (IRMS, Thermo Delta V IRMS, Thermo Scientific, Bremen, Germany). The internal standard with a known δ$^{15}$N value (α-aminobutyric acid, -8.17‰ ± 0.03‰) in each sample was checked to determine the reproducibility of the isotope measurements. The analytical run was accepted when the differences between the δ$^{15}$N values of α-aminobutyric acid measured by GC-IRMS and its true values were at most 1.5 ‰. The analytical precisions (SD, n=9) of the δ$^{15}$N measurements for derivatized amino acid standards ranged from 0.5‰ to 1.4‰. The difference between amino acid δ$^{15}$N values measured by using an elemental analysis/IRMS (EA/IRMS) and GC/MS/IRMS after empirical correction ranged from 0.1‰ to 1.3‰. Each reported value is the mean of at least three δ$^{15}$N determinations. The detailed instrument conditions and method validation were described in our previous study (Zhu et al., 2018, 2020b). Since asparagine and glutamine are converted to aspartic acid and glutamic acid in the hydrolysis process, respectively, the concentration and δ$^{15}$N value of combined aspartic acid represents the sum of aspartic acid and asparagine. The concentration and δ$^{15}$N value of combined glutamic acid represents the sum of glutamic acid and glutamine.

**2.3 Analysis of the dry deposition fluxes**

In this study, the atmospheric deposition fluxes of CAAs (mg N m$^{-2}$ d$^{-1}$) at four sampling sites are estimated from equation 1:

$$F_{dry} = C \cdot V_d \qquad (1)$$

where C is the atmospheric concentration (nmol N m$^{-3}$) and Vd is the dry deposition velocity (m s$^{-1}$). Vd is controlled by the aerosol size, wind speed, aerosol hygroscopicity, relative humidity and underlying surface.

Primary biological aerosols have been found to be distributed from nanometers up to about a tenth of a millimeter, with their size distribution influenced by their sources (Fröhlich-Nowoisky et al., 2016).

However, we did not obtain data on the size distribution of CAAs. In this study, the deposition velocities of protein-N were assumed to be the same as those used to estimate water-soluble nitrogen (WSON) dry deposition (0.012 m s$^{-1}$), given that protein-N is a significant component of WSON in aerosols and WSON has also been detected in both coarse and fine fractions (Zamora et al., 2011; Zhang and Anastasio, 2003). The uncertainty in the value for the dry deposition velocity can lead to the uncertainty in dry flux estimates. For particles in the size range where gravitational setting is the controlling factor, Vd values obtained by model and field experiment were consistent (Spokes et al., 2000). Duce et al. (1991) reported that under wind speeds ranging from 0 to13 m s$^{-1}$ and relative humidity between 0% and 100%, the deposition velocity for submicrometer aerosol particles is 0.1 cm s$^{-1}$ ± a factor of 3, while the deposition velocity for supermicrometer crustal particles is 1 cm s$^{-1}$, also with an uncertainty factor of 3. During the dust period, the wind speeds in Beijing, Tianjin, Shijiazhuang, and Taiyuan were 2.0~2.4 m s$^{-1}$, 4.1~6.1 m s$^{-1}$, 1.7~2.6 m s$^{-1}$ and 4.6~7.0 m s$^{-1}$, respectively, and the relative humidity were 25.8~28.4%, 24.3~28.0%, 24.3~29.1%, and 37.0~38.5%, respectively. The variations in wind speed and relative humidity across the four sampling cities were relative minor, and their ranges fell within those reported by Duce et al. (1991). Based on this, the uncertainty for the deposition velocity of aerosol protein-N in this study was set to a factor of 3. The detailed calculation method for the uncertainty range of protein-N deposition fluxes is provided in Supplementary Material, Text 1.

A "new" input of CAA-N (protein-N) supplied by the Gobi Desert for the ecosystems in the downwind region (Input Fdry) can be calculated from equation 2:

Input Fdry= Fdry·f           (2)

where Fdry is the atmospheric deposition fluxes of protein-N, which is calculated from equation 1.f is the contribution of the Gobi dust source at each sampling site, which is obtained from the nitrogen isotopic mass balance (equation 3). Uncertainties associated with f were derived from the $\delta^{15}$N variabilities of combined Gly and Leu in Gobi dust source (mean ± SD).

$\delta^{15}N_{AA \text{ during the dust period}} = \delta^{15}N_{AA \text{ during the non-dust period}} \cdot (1-f) + \delta^{15}N_{AA \text{ dust source}} \cdot f$   (3)

where $\delta^{15}N_{AA \text{ during the dust period}}$ is $\delta^{15}$N value of the specific amino acid measured during the dust period, $\delta^{15}N_{AA \text{ during the non-dust period}}$ is $\delta^{15}$N value of the specific amino acid measured during the non-dust period and $\delta^{15}N_{AA \text{ dust source}}$ is $\delta^{15}$N value of the specific amino acid measured in the Gobi dust source. f is the contribution of the Gobi dust source at each sampling site.

## 2.4 Auxiliary Data

The meteorological data during the sampling period were collected from the Global Weather and Climate Information Network (http://www.weatherandclimate.info/). The MODIS satellite images were available at the following website: https://worldview.earthdata.nasa.gov/.

## 2.5 Statistics

Statistical analysis was conducted by Origin 2018 (OriginLab Corporation, USA). We performed a one-way analysis of variance (ANOVA) for the concentrations of $PM_{10}$, the ratios of Ala%/Pro%, Gly%/Pro% and Glu%/Pro%, $\delta^{15}N$ values of Glycine and leucine, testing the effect of Asian dust events. Tukey's honestly significant difference (Tukey-HSD) test was used to compare the significant difference. Further,

the differences in the $\delta^{15}N$ values of combined glycine leucine, isoleucine, alanine and valine in $PM_{2.5}$ at four cities during the non-dust period and their corresponding values in the surface soil in the Gobi Desert were examined using the one-way ANOVA procedure and compared using the Tukey-HSD test. For all tests, statistically significant differences were considered at $p < 0.05$.

## 3 Results

### 3.1 Concentrations of $PM_{2.5}$ and $PM_{10}$ in Northern China

Figure 1 illustrated the variations in atmospheric $PM_{2.5}$ and $PM_{10}$ concentrations across Beijing (BJ), Tianjin (TJ), Shijiazhuang (SJZ), and Taiyuan (TY) throughout the sampling period. Periods when mass concentrations of $PM_{10}$ exceeded 500 μg m$^{-3}$ were as follows: in Beijing from 06:00 to 19:00 on March 28; in Tianjin from 09:00 to 13:00 on the same day; in Shijiazhuang from 11:00 to 18:00; and in Taiyuan

from 21:00 on March 28 to 03:00 on March 29, as indicated by a yellow shadow in Figure 1. A MODIS satellite image from NASA (https://worldview.earthdata.nasa.gov/) revealed a dust episode on March 28, 2018, with brown dust plumes visibly overlaying the North China Plain, and significant dust presence persisting on March 29, suggesting that the dust particles did not settle. Consequently, the filter samples from March 28 and 29 were categorized as the dust period, highlighted in yellow in Figure 1. The

remaining days of the sampling period were classified as the non-dust period.

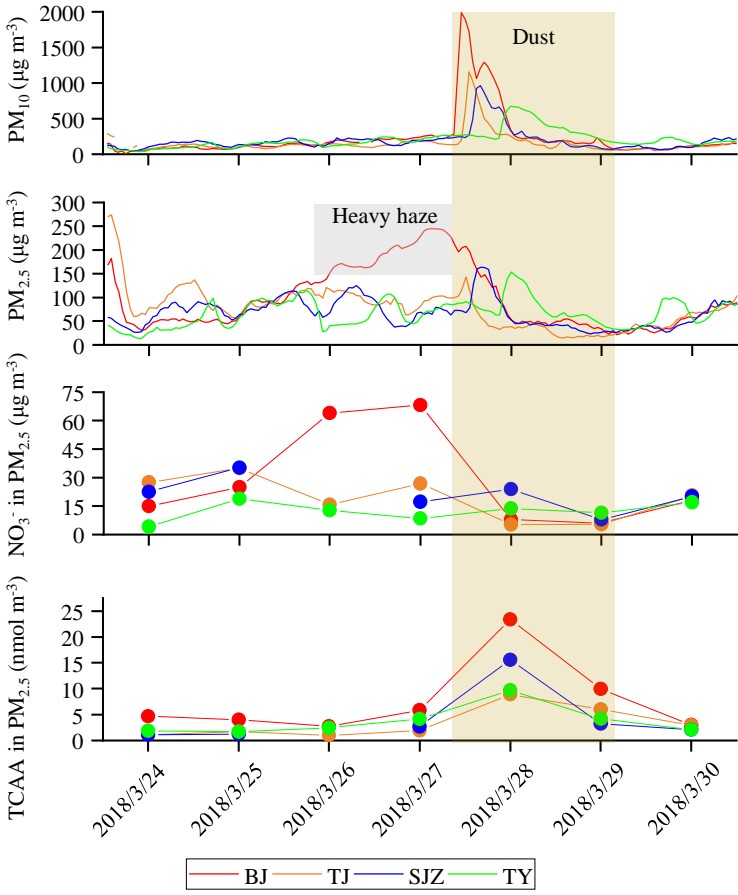

**Figure 1. Time series of PM$_{2.5}$ and PM$_{10}$ concentration, as well as NO$_3^-$, total combined amino acids (TCAA) in PM$_{2.5}$ at Beijing (BJ), Tianjin (TJ), Shijiazhuang (SJZ) and Taiyuan (TY) from March 24 to March 30, 2018. The timestamps indicate 21:00. The yellow shadow indicates the dust period. The grey shadow denotes a heavy haze period in BJ.**

As shown in Figure 1, compared to the non-dust period, the concentrations of PM$_{10}$ during the dust period exhibited a significant increase at four sampling sites, by factors of 8.7 in Beijing, 6.3 in Tianjin, 4.8 in Shijiazhuang, and 3.2 in Taiyuan ($p < 0.01$). Among four sampling sites, Beijing recorded the highest increase in PM$_{10}$ concentration with the peak value of 1989 μg m$^{-3}$ on 28 March (Figure 1). In comparison to PM$_{10}$, the increase in PM$_{2.5}$ concentration during the dust period was less pronounced. Beijing experienced a severe haze event two days before the dust invasion (March 26 and 27), with PM$_{2.5}$ concentrations surpassing 150 μg m$^{-3}$. Following the dust invasion, Beijing's PM$_{2.5}$ rapidly dropped to 50 μg m$^{-3}$ by 23:00 on March 28. In Tianjin, the PM$_{2.5}$ concentration first increased and peaked at 143 μg m$^{-3}$ at 08:00 on March 28 (dusty day). Subsequently, similar to Beijing, Tianjin's PM$_{2.5}$ rapidly decreased to 36 μg m$^{-3}$. Contrast to Beijing and Tianjin, PM$_{2.5}$ in Shijiazhuang and Taiyuan, showed an upward trend following the dust invasion, with daily averages rising from 62 and 78 μg m$^{-3}$ on March 27

to 79 and 94 μg m$^{-3}$ on March 28, respectively. Overall, there was a decrease in the PM$_{2.5}$/PM$_{10}$ ratio at all locations. This ratio declined from 0.7 to 0.1 in Beijing, 0.6 to 0.1 in Tianjin, 0.4 to 0.2 in Shijiazhuang, and 0.4 to 0.2 in Taiyuan (Table S3), further indicating a significant increase in coarse particulate matter

due to the Asian dust. The 48-h backward trajectories and clusters of air mass trajectories at the four sampling sites on March 28th, 2018 are displayed in Figure 2. The air masses arriving at the four sampling sites during the dust period primarily originated from the Gobi Desert. Observation from the MODIS satellite also confirmed that the occurrences of Asian dust events across four sampling sites originated from the Gobi Desert (Figure S1). Tian et al. (2020) also confirmed that an intensive dust event occurred

in the North China Plain from 26–29 March 2018 originating from western Inner Mongolia.

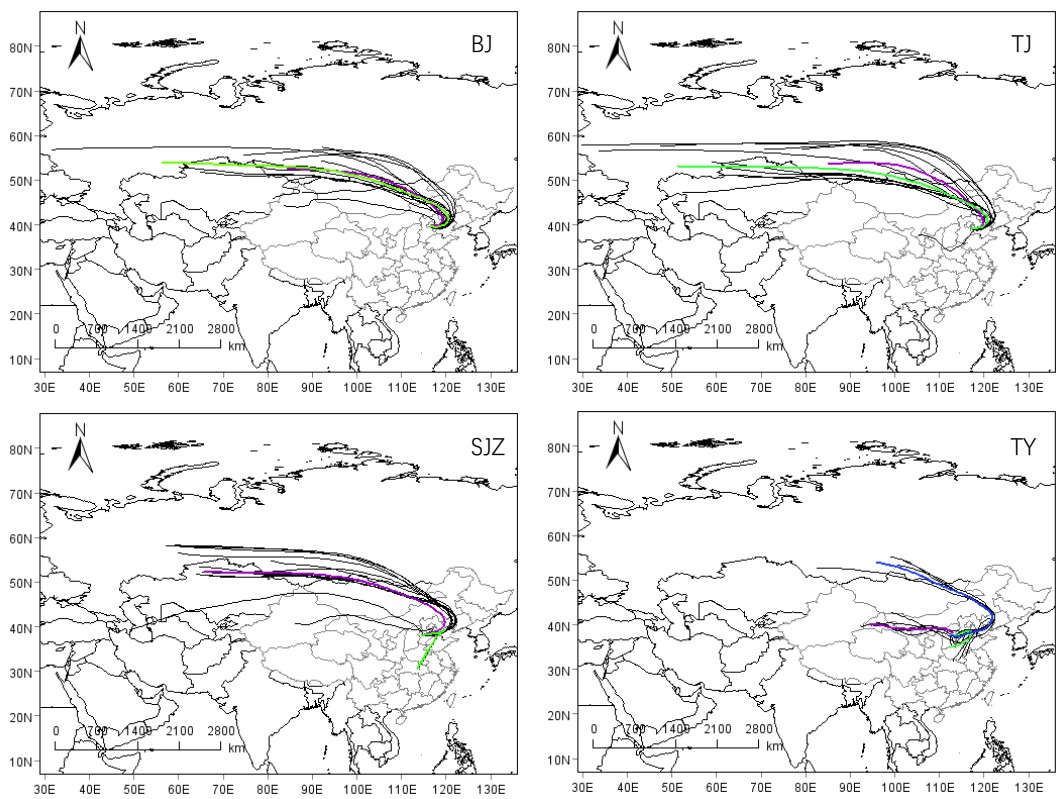

**Figure 2. 48h backward trajectories of air masses arriving in reaching Beijing, Tianjin, Shijiazhuang and Taiyuan at 500m above ground level during the dust period. Color lines show the cluster results of air mass trajectories. Trajectories were divided into two or three clusters in four cities. The main contribution clusters**

**in these cities were found to be primarily originated from the Gobi Desert.**

**3.2 Concentrations and distribution of CAAs**

**3.2.1 CAAs in the Gobi Desert sources**

Given that this Asian dust event originated from the Gobi Desert (Figure 2), we investigated the molar composition of CAAs in both the surface soils and dominant plants of the region to determine the characteristic molar percentages of proteins transported from the Gobi Desert. In the surface soil of the Gobi Desert, glycine and alanine were the most abundant combined amino acids, each contributing approximately $20.4 \pm 5.6\%$ and $20.7 \pm 2.9\%$, respectively, to the total CAAs pool (Figure 3a). Glutamic acid was the predominant CAA species in typical Gobi plants, accounting for 21.0%, 22.5%, and 69.3% of the total CAAs pool in *chenopodium ambrosioides*, *Tripogon chinensis*, and *tamarix chinensis*, respectively (Figure 3a).



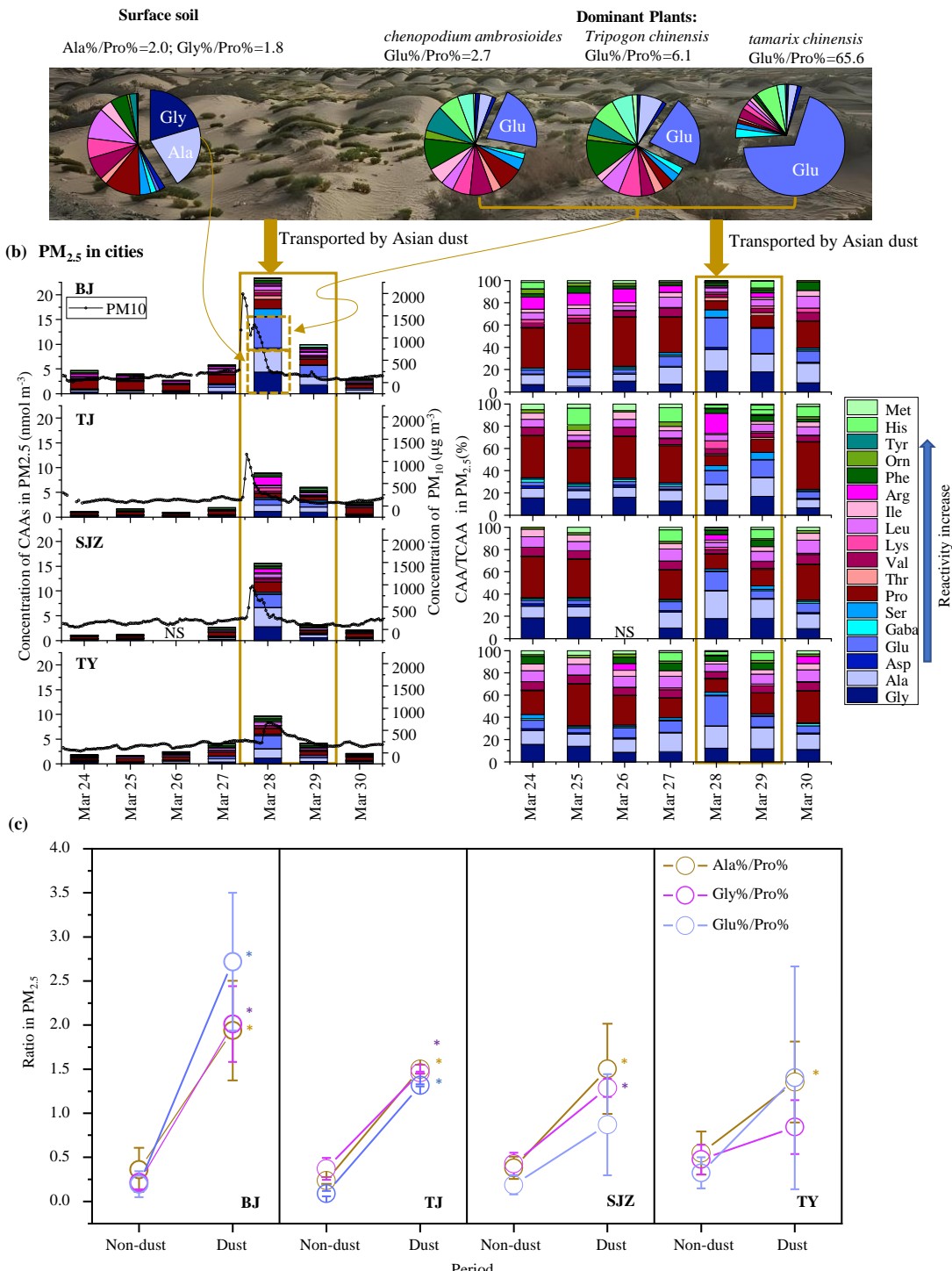

**Figure 3. (a)** The percent distributions of each individual CAAs (% of total CAAs) in the surface soils and three dominant plants (*chenopodium ambrosioides*, *Tripogon chinensis*, and *tamarix chinensis*) in the Gobi Desert. **(b)** Concentrations and distribution of individual CAAs at BJ, TJ, SJZ and TY. **(c)** the average ratios of the percentage of alanine, glycine, and glutamic acid to proline (Ala%/Pro%, Gly%/Pro%, Glu%/Pro%) in $PM_{2.5}$ on dusty and non-dust days at four cities. Asterisks indicate a significant difference in the average ratio between dust and non-dust periods (one-way ANOVA, $p < 0.05$).

In both the surface soil and common plants of the Gobi Desert, the molar percentage of combined proline was lower than those of the amino acids mentioned earlier, contributing only $11.4 \pm 3.3\%$ to the total

CAAs pool of the Gobi surface soil. Among the three common plants, the molar percentage of proline did not exceed 8%. Therefore, the proteins transported from Gobi Desert are predominantly composed of alanine, glycine, and glutamic acid (Figure 3a).

### 3.2.2 Total CAAs concentration in PM$_{2.5}$ in Northern China

The temporal variations of total CAAs in PM$_{2.5}$ measured at all sites during the sampling period were

shown in Figure 1. A marked increase in the total CAAs in fine particles were observed at all sampling locations. The average concentrations of total CAAs in PM$_{2.5}$ in Beijing, Tianjin, Shijiazhuang, and Taiyuan increased from the non-dust period of 4.0, 1.7, 1.8, and 2.5 nmol m$^{-3}$ to 16.7, 7.5, 9.4, and 6.9 nmol m$^{-3}$ during the dust period, respectively ($p < 0.01$). The temporal variation of the total CAAs concentration were consistent with that of PM$_{10}$ at four cities (Figure 1). The highest concentrations of

total CAAs at four sites occurred on 28 March 2018 coincided with peaks in the enrichment of PM$_{10}$ concentrations.

As exhibited in Figure 1, the temporal variation pattern of NO$_3^-$ concentration in PM$_{2.5}$ was similar to that of PM$_{2.5}$, which was quite different from that of total CAAs concentrations in PM$_{2.5}$ at Beijing and Tianjin. At Beijing and Tianjin, the highest concentration of NO$_3^-$ in PM$_{2.5}$ occurred on 27 March (non-

dust day), but decreased on 28 March and 29 March (dust day). The difference in the temporal variations of total CAAs and NO$_3^-$ in PM$_{2.5}$ may point to the contribution of different sources during the sampling period.

### 3.2.3 Individual CAAs in PM$_{2.5}$ in Northern China

For individual amino acid species in PM$_{2.5}$, the concentrations of alanine and glycine, which predominate

in the surface soil of the Gobi Desert, increased markedly during the dust period (Figure 3b). Specifically, alanine concentrations in PM$_{2.5}$ during the dust period were $3.0 \pm 2.0$, $1.1 \pm 0.2$, $2.2 \pm 2.3$ and $1.3 \pm 0.8$ nmol m$^{-3}$ at Beijing, Tianjin, Shijiazhuang, and Taiyuan, being 7, 8, 10 and 4 times more than that during the non-dust period, respectively. Glycine concentrations in PM$_{2.5}$ rose from $0.3 \pm 0.1$, $0.2 \pm 0.04$, $0.2 \pm 0.03$ and $0.3 \pm 0.07$ nmol m$^{-3}$ during the non-dust period to $3.1 \pm 1.8$, $1.1 \pm 0.1$, $1.7 \pm 1.6$ and $0.8 \pm 0.5$

nmol m$^{-3}$ during the dust period at Beijing, Tianjin, Shijiazhuang and Taiyuan, respectively. Similarly,

the concentrations of glutamic acid, which predominate in common plants of the Gobi Desert, increased markedly during the dust period, with concentrations 17, 15, 12 and 8 times higher than that of the non-dust period at Beijing, Tianjin, Shijiazhuang and Taiyuan, respectively (Figure 3b). In contrast, the concentrations of other CAAs displayed no significant variation between the dust and non-dust periods.

During the dust period, the molar composition of CAAs in $PM_{2.5}$ shifted significantly at all sampling locations, compared to the non-dust period (Figure 3b). In the non-dust period, combined proline dominated the total CAAs pool in $PM_{2.5}$, contributing to 35.4%, 36.3%, 32.3%, and 26.0% in Beijing, Tianjin, Shijiazhuang, and Taiyuan, respectively, as illustrated by the dark red shadow in Figure 2b. This prevalence diminished across all sites during the dust period, with molar percentages of combined proline

dropping to 9.4%, 10.3%, 13.9%, and 14.9%, correspondingly. Concurrently, there was a marked increase in the molar percentages of combined alanine, glycine, and glutamic acid (blue shadow in Figure 3b). The average molar percentages of these three CAAs in fine particulates increased from 24.4%, 24.9%, 31.6%, and 31.9%, respectively, during the non-dust period to 61.1%, 44.2%, 50.4%, and 48.8% during the dust period, thus emerging as the dominant CAA species at these locations. Notably, in Beijing,

the molar percentage of combined proline exhibited the largest decrease, while the sum of the molar percentage of combined alanine, glycine, and glutamic acid which is abundant in soil and plants of the Gobi Desert, showed the greatest increase.

Based on the variation trends in the percentage composition of CAAs in $PM_{2.5}$ across four cities during the dust and non-dust period, the ratios of the percentage of alanine, glycine, and glutamic acid to proline

(Ala%/Pro%, Gly%/Pro%, Glu%/Pro%) in $PM_{2.5}$ on dusty days were compared to those on non-dust days across the four cities. The results showed that in Beijing and Tianjin, the ratios of Ala%/Pro%, Gly%/Pro% and Glu%/Pro% increased significantly on dusty days compared to non-dust days ($p < 0.05$) (Figure 3c). In Shijiazhuang, during the dust period, the ratios of Ala%/Pro% and Gly%/Pro% also showed significant increases relative to non-dust days ($p < 0.05$), whereas in Taiyuan, only the Ala%/Pro% ratio increased

significantly during the dust period ($p < 0.05$) (Figure 3c).

### 3.3 Compound-specific nitrogen isotopes of CAAs ($\delta^{15}$N-CAAs)

### 3.3.1 $\delta^{15}$N-CAAs in Gobi Desert

$\delta^{15}$N-CAAs of individual CAAs in the surface soil and predominant plants in the Gobi Desert were compared with those in $PM_{2.5}$ at four urban sites during non-dust period (Figure 4, right side). Glycine, leucine, isoleucine, alanine and valine were $^{15}$N-depleted in the Gobi Desert surface soil compared to those in $PM_{2.5}$ at the urban sites during the non-dust period, whereas $\delta^{15}$N values of other individual CAA species were close between these two sources (Figure 4, right side). Among these four CAA species, the most significant $\delta^{15}$N depletion was observed for glycine and leucine. The mean $\delta^{15}$N values of glycine and leucine showed statistically significant differences between the Gobi Desert surface soil and urban $PM_{2.5}$ during the non-dust period (one way ANOVA, $p < 0.01$) (Figure 5). Specifically, glycine and leucine in Gobi Desert surface soil were depleted in $^{15}$N by 12 to 14‰ and 6 to 11‰, respectively, relative to their corresponding values in urban $PM_{2.5}$ during the non-dust period (Figure 4, right side, yellow box).

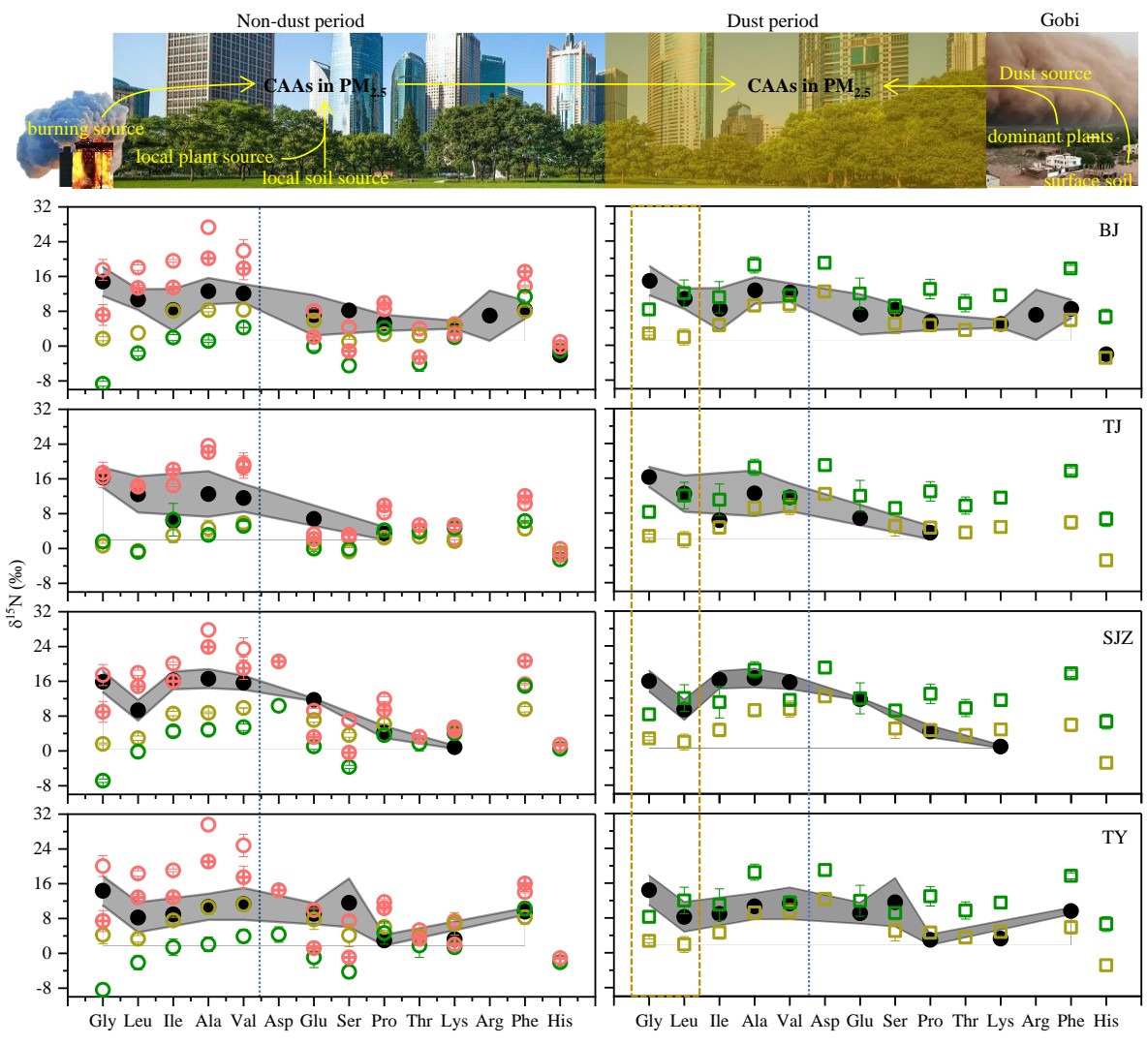

**Figure 4.** Comparison of $\delta^{15}N$-CAA patterns of in PM$_{2.5}$ at BJ, TJ, SJZ and TY with those in potential local sources and Gobi dust sources. CAAs species on the left side of blue dotted line were $^{15}N$-depleted in the Gobi Desert surface soil compared to those in PM$_{2.5}$ at the urban sites during the non-dust period. CAAs species in the yellow dotted box exhibited the most significant $\delta^{15}N$ depletion compared to those in PM$_{2.5}$ at the urban sites during the non-dust period.

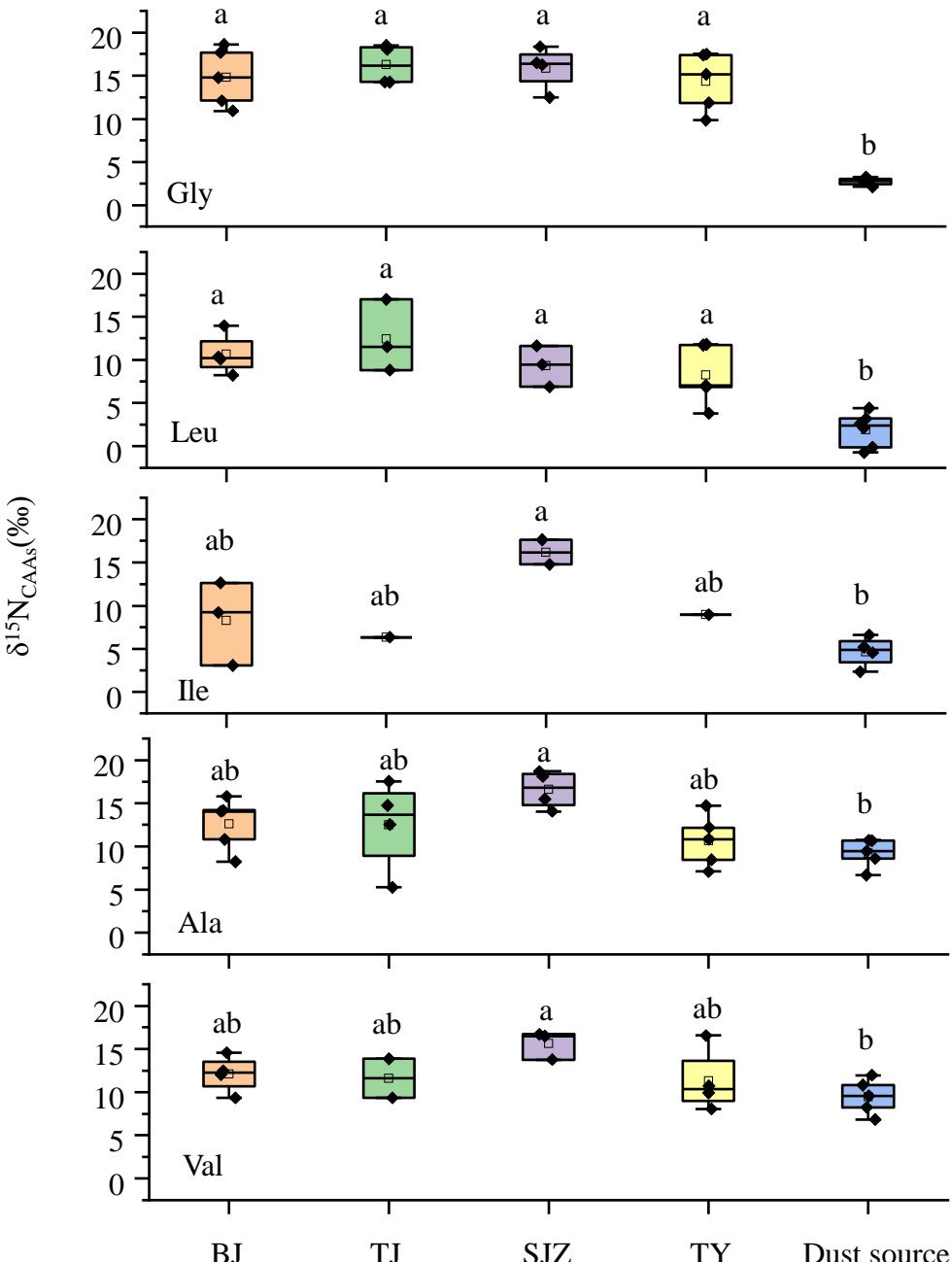

**Figure 5. The δ15N values of combined glycine (Gly), leucine (Leu), isoleucine (Ile), alanine (Ala) and valine (Val) in PM2.5 at Beijing (BJ), Tianjin (TJ), Shijiazhuang (SJZ) and Taiyuan (TY) during the non-dust period and their corresponding values in the surface soil in the Gobi Desert. Different lower-case letter denote means found to be statistically different (one-way ANOVA, $p < 0.05$).**

Compared to the surface soil of the Gobi Desert, predominant plants in the region exhibited significant [15]N enrichments of 2–12‰ for all CAA species (Figure 4, right side, green box). It is noteworthy that the average δ15N value of combined glutamic acid (11.9 ± 3.6‰), the predominant amino acid in these plants,

was close to its values in urban $PM_{2.5}$ during the non-dust period at four cities (one-way ANOVA, p > 0.05).

### 3.3.2 $\delta^{15}N$-CAAs in $PM_{2.5}$ in Northern China

Figure 4 shows the $\delta^{15}N$ patterns of individual CAAs in $PM_{2.5}$ (dark circle), local common plant (green circle) and local soil (yellow circle) as well as local burning sources (pink circle) from four urban sites. During the non-dust period, $\delta^{15}N$-CAAs patterns in $PM_{2.5}$ at four urban sites were generally consistent, with glycine, leucine, isoleucine, alanine and valine exhibiting relatively higher $\delta^{15}N$ values than other CAA species (Figure 4, left side). Besides that, $\delta^{15}N$ values of individual CAAs in $PM_{2.5}$ all fell within

their respective ranges observed in local dominant plants, soil, and burning sources at four cities (Figure 4, left side).

During the dust period, glycine, alanine, valine, leucine, and isoleucine showed negative shifts in $\delta^{15}N$ values compared to those observed during the non-dust periods (Figure 6). In contrast, the $\delta^{15}N$ shifts in other CAA species between the dust and non-dust periods were either relatively minimal or displayed

$\delta^{15}N$ enrichments (Figure 6). A spatial variation in $\delta^{15}N$ depletions for these five amino acids was observed, with the largest depletions occurring in Beijing. The degree of $\delta^{15}N$-CAA shifts decreased in the order of Beijing, Tianjin, Shijiazhuang, and Taiyuan. In Taiyuan, the $\delta^{15}N$ shifts in all five amino acids during the dust period were not observed.

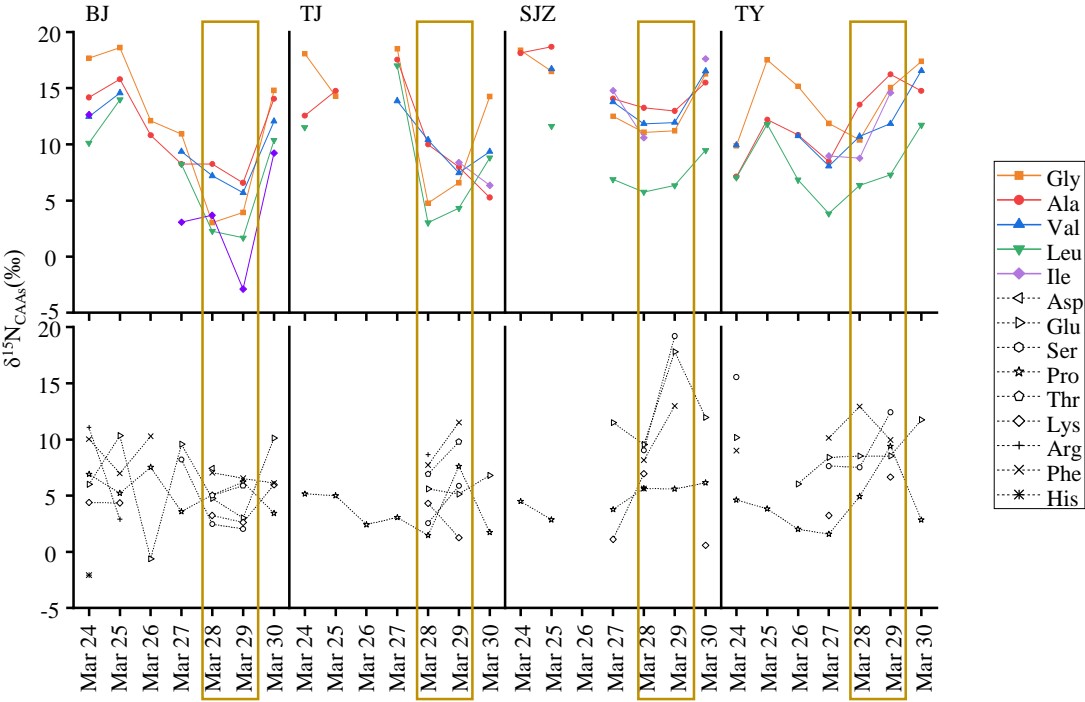

**Figure 6. Time series of δ¹⁵N of individual CAAs in PM₂.₅ at BJ, TJ, SJZ and TY. The yellow box represents the dust period.**

Among these five amino acids, glycine and leucine in PM₂.₅ exhibited the most significant δ¹⁵N depletion during the dust period at four sampling sites (Figure 6). The δ¹⁵N values of combined glycine in PM₂.₅ significantly decreased from 14.8‰, 16.3‰, and 15.9‰ during the non-dust period to 3.5‰, 5.7‰ and 11.1‰, respectively, during the dust period at Beijing, Tianjin and Shijiazhuang (one-way ANOVA, p < 0.05) (Figure 7a). At Taiyuan, the average δ¹⁵N values of combined glycine in PM₂.₅ during the non-dust (14.4‰) and dust period (12.7‰) were not significantly different (p > 0.05). Remarkably, the δ¹⁵N values of combined glycine in Beijing's PM₂.₅ during the dust period ranged from 3.0‰ to 3.9‰, closely aligning with those found in surface soils of the Gobi Desert, which ranged from +2.1‰ to +3.2‰ (Figure 7a). Similarly, the δ¹⁵N values of combined leucine in PM₂.₅ from Beijing and Tianjin sharply decreased from +10.7‰ and +12.4‰ during the non-dust period to +2.0‰ and +3.7‰, respectively, during the dust period (one-way ANOVA, p < 0.05) (Figure 7b). At Shijiazhuang and Taiyuan, there was no significant variation in the mean δ¹⁵N values of leucine between the non-dust and dust period (p>0.05) (Figure 7b).

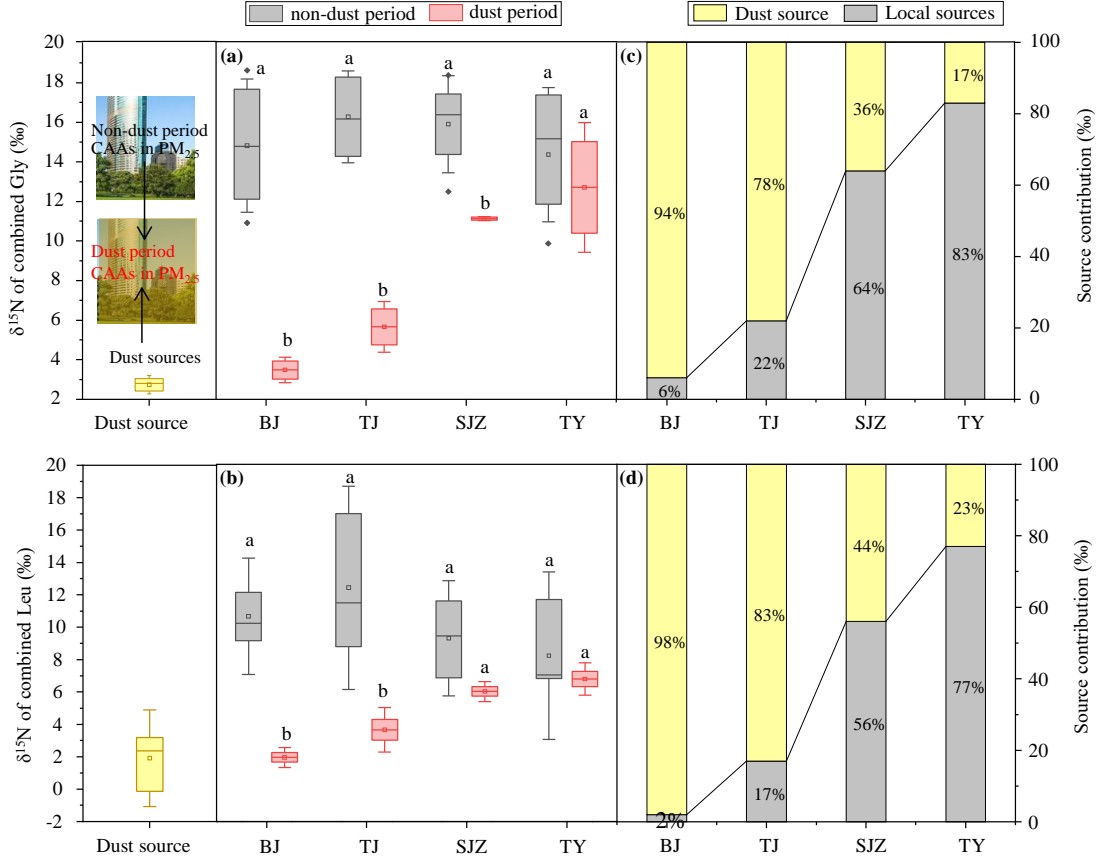


**Figure 7. (a) $\delta^{15}$N values of combined glycine in surface soil from Gobi Desert (yellow box) and in PM$_{2.5}$ during the non-dust (grey box) and dust (red box) period at Beijing, Tianjin, Shijiazhuang and Taiyuan; (b) $\delta^{15}$N values of combined leucine in surface soil from Gobi Desert (yellow box) and in PM$_{2.5}$ during the non-dust (grey box) and dust (red box) period; (c) the contribution of Gobi dust sources and local urban sources to the**

**CAAs in PM$_{2.5}$ calculated by the nitrogen isotopic mass balance for glycine; (d) the contribution of Gobi dust sources and local urban sources to the CAAs in PM$_{2.5}$ calculated by the nitrogen isotopic mass balance for leucine. Different lower-case letter denote means found to be statistically different (one-way ANOVA, p < 0.05) between the non-dust and dust period.**

Moreover, as mentioned above, one-way ANOVA revealed significant differences in the $\delta^{15}$N values of

glycine and leucine between Gobi Desert surface soil and urban PM$_{2.5}$ during the non-dust period (p<0.01; Figure 5). Therefore, a simple nitrogen isotopic mass balance was employed to estimate the contribution of long-range transported Gobi dust sources to proteinaceous matter in PM$_{2.5}$ at four cities. The results of the nitrogen isotopic mass balance for both glycine and leucine were consistent. During the dust period, long-range transported Gobi dust sources contributed $94 \pm 17\%$ ~ $98 \pm 23\%$, $78 \pm 7\%$ ~ $83 \pm 11\%$, $36 \pm$

$1\%$ ~ $44 \pm 12\%$ and $17 \pm 25\%$ ~ $23 \pm 10\%$ to proteinaceous matter in PM$_{2.5}$ at Beijing, Tianjin, Shijiazhuang and Taiyuan, respectively (Figure 7c and d).

### 3.4 Dry deposition fluxes of protein-N

A "new" input of CAA-N (protein-N) supplied by the Gobi Desert for the ecosystems in the downwind region were calculated from equation 2. The contribution of the Gobi dust source at each sampling site (f) was obtained from the nitrogen isotopic mass balance (Figure 7c and d). The average dry deposition of protein-N collected at each site during the non-dust period served as the background levels. The dry deposition flux of protein-N in $PM_{2.5}$ sharply increased from the non-dust to the dust period at all four sampling sites. During the dust period, the dry deposition of protein-N (Fdry) in $PM_{2.5}$ was 0.37, 0.22, 0.28, and 0.15 mg N $m^{-2}$ $d^{-1}$ in Beijing, Tianjin, Shijiazhuang and Taiyuan, respectively. The input of protein-N from the Gobi dust (Input Fdry) was approximately 4.5, 6.3, 3.3, and 1.3 times higher than the background levels in Beijing, Tianjin, Shijiazhuang, and Taiyuan, respectively (Figure 8).

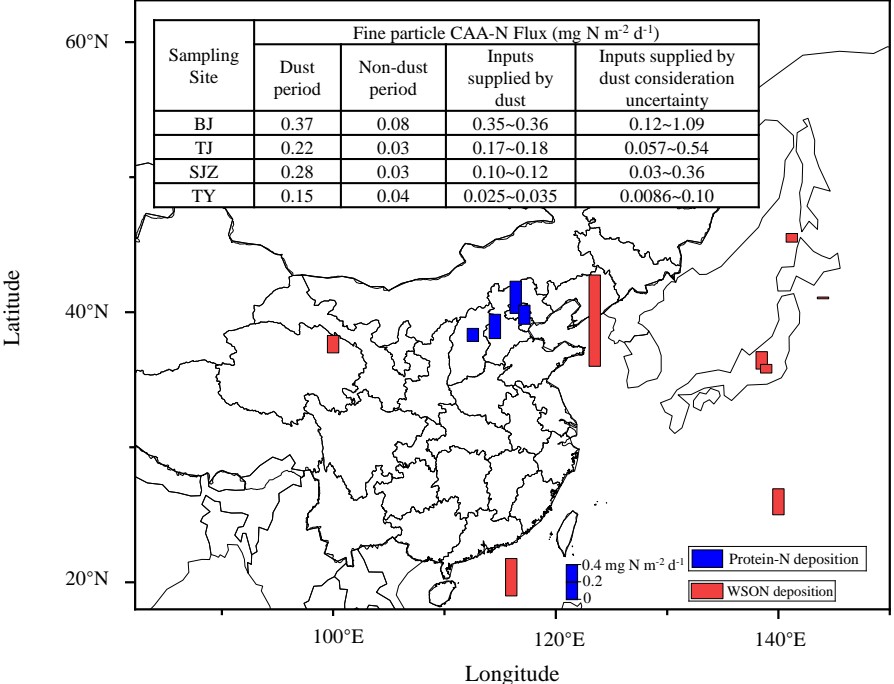

**Figure 8. The dry deposition flux of protein-N at BJ, TJ, SJZ and TY during the dust period (blue bar) and published reports of the dry deposition flux of WSON measured in different atmospheric scenarios (red bar). The length of the bars represents the dry deposition flux of WSON and protein-N. The WSON deposition data were sourced from previous studies (Shi et al., 2010; Ho et al., 2015; Matsumoto et al., 2014; Tsagkaraki et al., 2021; Nakamura et al., 2006; Zhang et al., 2011). The method of calculating deposition fluxes of protein-**

 **N with the uncertainty range at the four sampling sites was provided in the supplementary manuscript (Text 1).**

## 4 Discussion

### 4.1 Local urban sources of CAAs in PM$_{2.5}$

Amino acids constitute a major component of water-soluble organic nitrogen and play significant roles in biological activities. Previous studies focused on the levels and distribution of free amino acids in urban aerosols during the dust period (Mace et al., 2003; Shi et al., 2010), revealing that free amino acids typically represent a minor portion of the organic nitrogen, accounting for approximately 1% of the total organic nitrogen, with no significant increase in their average concentrations during dust periods. However, CAAs are the predominant form of the amino acid compounds in aerosols, with concentrations approximately five times higher than those of FAAs (Matsumoto et al., 2021; Wedyan and Preston, 2008). Limited studies investigated that the variation of the sources of CAAs in urban aerosols from the non-dust to dust period.

Primary biological aerosol particles including bacteria, fungal spores, viruses, algae, pollen, biological crusts, and plant or animal fragments and detritus, are considered the major sources of CAAs in aerosols (Després et al., 2012; Matos et al., 2016; Xie et al., 2024). They are not only emitted from natural sources but also are associated with anthropogenic activities such as soil resuspension from traffic and construction, industry, agricultural practices, wastewater treatment and biomass burning (Kang et al., 2012; Matos et al., 2016; Song et al., 2017). During non-dust period, CAAs in aerosols at four urbans were mainly influenced by the local sources. The average concentrations of total CAAs in PM$_{2.5}$ in Beijing, Tianjin, Shijiazhuang, and Taiyuan during non-dust period were 4.0, 1.7, 1.8, and 2.5 nmol m$^{-3}$, respectively. Based on the molecular weight of amino acids, the molar concentrations were converted to mass concentrations, which were 0.5, 0.2, 0.2 and 0.3 μg m$^{-3}$, respectively. These results are similar to levels previously measured in urban Beijing (average: 0.65±0.55 μg m$^{-3}$) (Wang et al., 2019), and urban Nanchang, China (average: 0.3±0.3 μg m$^{-3}$) (Zhu et al., 2020b), but higher than that observed in rural Guangzhou, southern China (average: 0.13±0.05 μg m$^{-3}$) (Song et al., 2017) and a coastal site of Okinawa, Japan (average: 0.16±0.10μg m$^{-3}$) (Li et al., 2022a). However, significantly higher average protein concentrations were observed in PM$_{2.5}$ in Xi'an during haze pollution periods (average: 5.46±3.32 μg m$^{-}$

$^3$) (Li et al., 2022b) and in $PM_{10}$ in Hefei (average: 11.42 g m$^{-3}$) (Kang et al., 2012). Clearly, cities with heavy traffic, agricultural production activities, construction activities, and various industries including machinery, electronics, chemistry, steel, textile, and cigarette manufacturing have higher atmospheric protein concentrations. Local road dust resuspension and anthropogenic activities in urban areas are important contributors to proteins in aerosols. Besides that, this study found a prevalent concentration of combined proline in the total CAAs pool of $PM_{2.5}$ across all sampling sites during the non-dust period. The compositional profiles of CAAs in aerosols have been utilized to identify the sources of primary aerosol particles (Abe et al., 2016; Matsumoto et al., 2021). The sampling occurred in spring, a period marked by rapid growth of plants and spore emissions. Proline has previously been reported as the major CAA species in urban plants and spores (Barbaro et al., 2015; Matsumoto et al., 2021; Zhu et al., 2020b). Therefore, the predominance of Pro among CAAs observed these four representative urbans in North China Plain points to the significant contribution of local plants and spores during spring.

As discussed above, local vegetation, road dust resuspension (traffic and construction activities), and anthropogenic industrial activities may be the major local sources of atmospheric CAAs. Therefore, we determined the nitrogen isotope values of individual CAAs in *Platanus orientalis*, the most typical plant of the North China Plain and commonly used for urban greening, to serve as the endpoint value for plant-emitted bioaerosols (green circle, Figure 4 left). The $\delta^{15}N$ values of individual CAAs measured in road dust sampled from each city (yellow circle, Figure 4 left) were used as the endpoint value for road dust resuspension. However, the $\delta^{15}N$ values of individual CAAs in $PM_{2.5}$ were more positive than those measured in local representative plant and road dust during the non-dust period, further confirming protein in $PM_{2.5}$ was significantly impacted by sources with $\delta^{15}N$ values of individual CAAs more positive than those of local plant and soil sources, particularly for glycine, leucine, isoleucine, alanine, and valine (Figure 4). Previous studies have suggested that anthropogenic industrial activities, such as the combustion of coal, biomass, or organic materials, contribute significantly to urban aerosol proteins, as indicated by the correlations between aerosol proteins and water-soluble ions (Khan et al., 2019; Li et al., 2022b). Thus, these sources with $^{15}N$ enrichment are likely anthropogenic industrial activities. Since CAAs are exposed to burning processes during these industrial activities, $\delta^{15}N$ values of CAAs emitted from these processes are influenced by nitrogen isotopic fractionation of each CAAs during the burning processes. Our earlier research showed that during combustion, the nitrogen isotope values of glycine,

leucine, isoleucine, alanine, and valine increased by 15.8‰, 15.0‰, 11.5‰, 19.0‰, and 13.6‰, respectively, compared to their initial values, while other amino acids showed an isotopic increase of less than 5.8‰ (Zhu et al., 2024). Clearly, the combustion process led to significant enrichment of nitrogen isotopes in specific amino acids, with the most substantial $^{15}N$ enrichment observed in glycine, leucine, isoleucine, alanine, and valine. We used $\delta^{15}N$ values of individual CAAs in local plant and road dust as initial values and the nitrogen isotopic fractionation of each CAA during the combustion process to calculate the $\delta^{15}N$ values of individual CAAs in local biological materials after undergoing combustion (pink circle, Figure 4 left). As shown in Figure 4, during the non-dust period, the $\delta^{15}N$ values of individual CAA in PM$_{2.5}$ from four cities all fell within their respective ranges in local common plants, soil, and burning sources across the four urban sites. This further supported that local dominant plants, surface road dust and anthropogenic industrial activities were the major sources of CAAs in PM$_{2.5}$ across Beijing, Tianjin, Shijiazhuang, and Taiyuan during the non-dust period.

**4.2 Identification of long-range transported dust source to the CAAs in PM$_{2.5}$**

Asian dust particles carry substantial microorganisms in desert soils and travel over long distances, leading to an increase in airborne microorganisms in remote downstream areas (Maki et al., 2017; Tang et al., 2018; Tong et al., 2023). Microbial particle concentrations in the atmosphere are proved to be positively related to coarse particles during the dust period (Hara and Zhang, 2012; Maki et al., 2019) (Puspitasari et al., 2016). The content of protein in the atmosphere is proved to be the combination of pollen, spore, bacteria, viruses, debris from humans, animals and plants, or fecal matter as well as nucleated combinations of these particles, which can serve as an indicator of all biological matter in the air (Boreson et al., 2004; Menetrez et al., 2007; Staton et al., 2015). It is reasonable to hypothesize that during the dust period, the proteinaceous matter in desert could be one of the potential sources of CAAs in aerosols in downwind cities.

This study presents the first simultaneous measurements of the levels and distribution of CAAs in PM$_{2.5}$ across four urban areas during a dust event. The concentrations of total CAAs in PM$_{2.5}$ increased sharply at all sampling sites during the dust period ($p < 0.01$) (Figure 1). Furthermore, the temporal variation pattern of total CAAs in PM$_{2.5}$ was consistent with that of PM$_{10}$ (Figure 1). An increase in PM$_{10}$ indicates a rise in dust coarse particle levels (Wu et al., 2019). Similarly, Xie et al. (2024) reported that the

concentration of aerosol proteins strongly correlated with the mass concentration of particles larger than

2.5 μm during the dust periods, indicating a close dependence of aerosol proteins on dust particles.

Furthermore, the total CAAs concentrations in $PM_{2.5}$ were not correlated with the local meteorological

conditions (including temperature, humidity or wind speed), but were positively correlated with $PM_{10}$

concentration during the dust period (Table S4). suggesting the increase concentration of CAAs in $PM_{2.5}$

during the dust period was not influenced by local meteorological conditions, but were directedly linked

to the long-transported Gobi dust source. Conversely, the concentration of total CAAs displayed an

inverse temporal variation pattern with nitrate ($NO_3^-$) concentration (Figure 1). $NO_3^-$ is a key component

of secondary inorganic aerosols in $PM_{2.5}$, which formed via photochemical oxidation reactions of NOx

(Song et al., 2021). Previous studies demonstrated that local urban coal combustion, vehicle exhausts,

biomass burning, and microbial N cycle were the major contributors to the atmospheric $NO_3^-$ in Beijing

(Song et al., 2019; Zhang et al., 2021). Therefore, decreasing trend of $NO_3^-$ concentration in Beijing and

Tianjin during the dust period suggested the lower contribution of local urban sources to CAAs in $PM_{2.5}$

at these cities during the dust period. During the dust period, the distribution profiles of CAAs in $PM_{2.5}$

at all observation sites changed, exhibiting consistent variation patterns. Notably, there was a significant

increase in the proportions of combined glycine, alanine and glutamic acid, the most abundant amino

acids in surface soil and predominant plants from the Gobi Desert (Figure 3). Moreover, substantial

amounts of alanine, glycine and glutamic acid from Gobi Desert surface soil and plants into $PM_{2.5}$ in

downwind areas led to increased ratios of Ala%/Pro%, Gly%/Pro%, and Glu%/Pro% in $PM_{2.5}$ at four

cities on dusty days (Figure 3). These findings further support the hypothesis that long transported surface

soil and plants from the Gobi Desert significantly contribute to the proteinaceous matter in $PM_{2.5}$ in

downwind areas during the dust period. These newly protein transported from the Gobi Desert area has

high proportions of combined glycine, alanine and glutamic acid, which is abundant in surface soil and

predominant plants in Gobi Desert.

The $\delta^{15}N$ values of specific amino acid have been utilized as a novel method to identify the sources of

amino acids (Batista et al., 2014; Mccarthy et al., 2013; Zhu et al., 2021). The shift in $\delta^{15}N$ values of

specific CAAs in urban $PM_{2.5}$ during the dust period may be attributed to the inputs from long-range

transport amino acid sources. Combined glycine constitutes a high proportion of the total CAAs pool in

the surface soil of the Gobi Desert (averaging 20.4%), while its content is extremely low in common

Gobi plants (averaging 1.1%) (Figure 3). Therefore, during dust periods, the $\delta^{15}N$ values of glycine in PM$_{2.5}$ in downwind cities are primarily influenced by the $\delta^{15}N$ values of glycine in the surface soil of the

Gobi Desert, rather than by those in Gobi plants. Since the mean $\delta^{15}N$ values of glycine in Gobi Desert surface soil was more negative than that of urban PM$_{2.5}$ during the non-dust period, the mean $\delta^{15}N$ values of glycine in PM$_{2.5}$ from cities impacted by Gobi dust sources would more negative. Leucine is present in roughly equal amounts in both the surface soil and the common plants of the Gobi Desert (averaging about 5.5% each) (Figure 3). The $\delta^{15}N$ values of leucine in predominant plants do not significantly differ

from those in PM$_{2.5}$ during the non-dust period, while $\delta^{15}N$ values of leucine in surface soil were significantly lower (Figure 4). Similar to glycine, during dust periods, the $\delta^{15}N$ values of leucine in PM$_{2.5}$ in downwind cities are also primarily influenced by the $\delta^{15}N$ values of leucine in the surface soil of the Gobi Desert. Therefore, the $^{15}N$ depletion of glycine and leucine in urban PM$_{2.5}$ indicated the contribution of Gobi dust source (Figure 5 and 7).

Combining the correlation between combined amino acids and PM$_{10}$, the variation in the distribution of the CAAs as well as the shift of the $\delta^{15}N$ values of individual amino acids in PM$_{2.5}$ during the dust period, we can conclude that CAAs in surface soil and predominant plants in the Gobi Desert are substantial contributors of combined amino acid in fine particles in Northern China during the dust period.

**4.3 Contribution of dust sources to the CAAs in PM2.5 at each city**

It is crucial to quantifying the contribution of dust sources to the CAAs in atmospheric PM$_{2.5}$ in different downwind regions. Although all four cities were affected by the same dust source (the Gobi Desert) during this dust event, the extent of increase in CAAs concentrations, the percentage increase in CAA species that are abundant in the Gobi Desert source (Ala%, Gly%, and Glu%), and the degree of $\delta^{15}N$ depletion in glycine and leucine varied among the four sampling sites (Figures 3 and 5). Notably, in

Beijing, the increase in concentration and percentage of alanine, glycine, and leucine, as well as the degree of $\delta^{15}N$ depletion in glycine and leucine, were the most significant. These results suggest that the contribution of dust sources to the CAAs in atmospheric PM$_{2.5}$ varies across the four sites, with Beijing, being closest to the dust source, most strongly affected by the Gobi dust sources. This result was consistent with the results of Xie et al. (2023). They examined the dissemination of bioaerosols in the

westerly wind from the Asian continent to the northwestern Pacific. The concentration of bioaerosol

increased significantly in all sites, which are located on the pathway of the Northern Hemisphere middle latitude westerly wind, during the dust period. However, increase rate of bioaerosols decreased with the distance from the Asian continent, suggesting the influence of dust on bioaerosol concentration along the transport pathway was in the decreasing order. They suggested that this decreasing trend was very likely

dominated by the dry deposition of dust particles (Xie et al., 2023, 2024). Dry deposition flux refers to the mass of particles in the atmosphere that settle to the surface by non-precipitation events (Xie et al., 2023). However, it is challenging to accurately estimate the contribution of protein originating from the dust sources and its dry deposition flux. Compound-specific nitrogen isotope analyses of individual amino acids may offer a new approach to solving this problem.

This study represents the isotopic interpretation of individual CAAs in $PM_{2.5}$ during the dust period to estimate the contribution of dust sources to CAAs in $PM_{2.5}$ in Northern China. As discussed in Section 4.2, significant quantities of proteinaceous matter transported by Asian dust are expected to increase the concentrations of combined glycine and leucine in urban $PM_{2.5}$ and decrease their $\delta^{15}N$ values. Greater inputs of proteinaceous matter from Asian dust typically result in more negative $\delta^{15}N$ values of combined

glycine and leucine in urban $PM_{2.5}$. Particularly in Beijing, $\delta^{15}N$ values of combined glycine and leucine in $PM_{2.5}$ during the dust period were close to those in the surface soil of the Gobi Desert (Figure 7). Therefore, with known concentrations and isotopic values of specific amino acid in $PM_{2.5}$ during the non-dust periods and during the dust period in downwind regions, as well as the isotopic values of these amino acids from the dust source area, it is possible to calculate the contribution of dust sources to the

protein in $PM_{2.5}$ in these downwind regions using nitrogen isotope mass balance for the specific amino acid. Using this approach, the contributions of dust sources to Beijing, Tianjin, Shijiazhuang, and Taiyuan during this dust event were calculated for glycine as $94 \pm 17\%$, $78 \pm 7\%$, $36 \pm 1\%$, and $17 \pm 25\%$ respectively. For leucine, the contributions were $98 \pm 23\%$, $83 \pm 11\%$, $44 \pm 12\%$, and $23 \pm 10\%$, respectively. The contributions estimated through nitrogen isotope mass balance for glycine align with those derived from leucine (Figure 7). This demonstrates that isotopic mass balance based on the $\delta^{15}N$

values of glycine and leucine in $PM_{2.5}$ is an effective tool for assessing the contribution of dust sources to proteinaceous material in downwind regions.

It should be noted that we applied a two-endmember mixing model comparing compound-specific nitrogen isotopic values of CAAs from Gobi dust sources with those in urban aerosols during non-dust

periods (representing the atmospheric background values), rather than those in local dominant plants, road dust and anthropogenic activities sources. Our methodology relies on the fundamental assumption that the nitrogen isotopic composition of CAAs in urban aerosols during the dust period, which derived from local dominant plants, road dust and anthropogenic activities, does not significantly differ from background atmospheric values.

For the Beijing, Tianjin and Shijiazhuang sampling sites, meteorological conditions (wind speed, relative humidity, and temperature) did not differ significantly ($p > 0.05$) between dust and non-dust periods (Figures S3 and S4). Under these stable conditions, local urban emission sources remained consistent, maintaining unchanged atmospheric background values. Therefore, the application of a two-endmember mixing model - utilizing isotopic values of compound-specific CAAs from Gobi dust sources and urban

aerosols during non-dust periods - provides a scientifically robust approach for quantifying Gobi dust contributions in Beijing, Tianjin and Shijiazhuang.

At the Taiyuan sampling site, wind speeds during dust events were significantly higher than during non-dust periods ($p < 0.05$; Figures S3–S4). These strong winds may enhance entrainment of local plant debris and road dust into aerosols, potentially modifying baseline $\delta^{15}N$ signatures of CAAs. This suggests

CAA $\delta^{15}N$ values during the non-dust period (atmospheric background values) may not fully represent local signatures during the dust event in Taiyuan, potentially affecting source apportionment. To evaluate this effect, we applied the MIXSIAR model (Stock and Semmens, 2016; Song et al., 2021) with $\delta^{15}N$ values of both glycine and leucine from local and the Gobi dust sources (Table S1, details were provided in Supplementary Materials, Text2). The MIXSIAR model showed that in Taiyuan, the relative

contributions of local dominant plants, road dust, and anthropogenic activities sources to aerosol CAAs averaged $1.6 \pm 2.6\%$, $46.2 \pm 20.9\%$ and $45.8 \pm 12.2\%$, respectively, during the non-dust period (Table S5). The dust period exhibited modified contribution profiles: local plants ($12.6 \pm 10.3\%$), road dust ($32.7 \pm 19.8\%$), and anthropogenic emissions ($35.9 \pm 10.0\%$), along with an external contribution from Gobi dust sources ($18.7 \pm 14.1\%$) (Table S5). The Gobi dust contribution estimated by MIXSIAR showed

agreement with our two-endmember model estimates ($17 \pm 25\% \sim 23 \pm 10\%$). After normalizing MIXSIAR results to exclude Gobi dust contributions, natural sources (local plant + road dust) showed only a modest increase from 47.8% (non-dust) to 55.8% (dust periods), representing an 8% enhancement. Future research should incorporate more extended observational periods of dust events, with particular

emphasis on downwind areas experiencing significant meteorological changes. Such extended investigations will enable more accurate assessment of how long-range transported dust sources influence biogeochemical cycles in downwind ecosystems.

Using this two-endmember mixing model, the calculated dry deposition of protein-N in $PM_{2.5}$ was 0.12~1.09, 0.057~0.54, 0.03~0.36, and 0.0086~0.10 mg N $m^{-2}$ $d^{-1}$ in Beijing, Tianjin, Shijiazhuang and Taiyuan, respectively, which displaying a decreasing trend along the transport pathway (Figure 2). Consistent trend was also found in the dry deposition flux of Asian dust, which decreased exponentially along its transport pathway (Lyu et al., 2017; Park et al., 2010). This result confirmed that the increase of airborne protein concentration in downwind areas during the dust period may be controlled by dry deposition of dust particles with protein coagulated or condensed.

### 4.4 Implications of the Gobi Desert-supplied protein-N inputs

Proteinaceous matter in the atmosphere has been studied extensively because it can be a utilizable source of nitrogen for plants and microorganisms (Ho et al., 2015; Samy et al., 2013; Zhang and Anastasio, 2003). Nutrients (e.g., protein-N) whipped up from deserts by strong winds can travel long distances, including over remote oceans (Favet et al., 2013; Yang et al., 2021). Previous studies have shown that dust inputs can stimulate phytoplankton metabolism and thereby enhance new production in the ocean (Duarte et al., 2006; Gazeau et al., 2021). The positive correlation of dust events with chlorophyll a concentrations, primary production and spring algae blooms in the south Yellow Sea and East China Sea has been well characterized (Tan et al., 2011). Due to the high bioavailability of amino acids-N, it can serve as a source of nutrients for marine ecosystems and greatly contribute to ecological processes in the marginal seas of China via processes such as fertilization of phytoplankton and stimulation of nitrogen fixation (Duarte et al., 2006; Ho et al., 2015).

In this study, the Gobi Desert-supplied protein-N input (0.12~1.09, 0.057~0.54, 0.03~0.36, and 0.0086~0.10mg N $m^{-2}$ $d^{-1}$ in Beijing, Taiyuan, Shijiazhuang and Taiyuan, respectively) (Figure 8) was compared with published reports of the dry deposition flux of WSON measured in different atmospheric scenarios. Gobi Desert-supplied protein-N input was higher than that observed in a suburban site in northern California, U.S.A. (0.04 mg N $m^{-2}$ $d^{-1}$) (Zhang et al., 2002). Although the Gobi Desert-supplied protein-N input was lower than the dry deposition of WSON measured in urban sites in China and a

coastal site in Qingdao, China, it was markedly higher than those previously measured in the remote Pacific Ocean and Atlantic Ocean areas, consistent with or even higher than those observed at the marginal seas of China (e.g., Yellow Sea and South China Sea) and forested, coastal and rural sites of Japan. Nakamura et al. (2006) suggested that WSON transported from East Asia is an important nitrogen component over the East China Sea. Our results showed that Gobi Desert-supplied protein-N inputs were comparable to or even higher than the dry deposition of WSON in the marginal seas of China (Figure 8). According to East Asian winter monsoon (from October to April) dynamics, a large quantity of protein-N supplied by Gobi dust found in this study can be deposited in a short period of the year. Although dust events happened within a few days, continuously occurring dust events and enhanced Asian dust supplied protein-N during the spring, coinciding with the algae bloom season, may exert widespread effects on the productivity of downwind ecosystems under the influence of the East Asian dust belt, including China, Korea, Japan and the North Pacific. Additionally, during long-range transport, hydrophilic protein particles mix with mineral particles to form various internal mixtures. These mixed particles can absorb water efficiently, thereby increasing CCN activity and altering optical properties, which is important for understanding global climate and hydrological cycles (Adachi et al., 2020). Furthermore, Li et al.(2022b) suggest that aerosol proteins might affect the generation of secondary aerosols. Under high relative humidity condition, water might condense onto protein aerosols due to their good hygroscopicity, resulting in forming a film of water on the aerosol surface. This water film is easy to adsorb gaseous pollutants in the atmosphere, including $SO_2$, NOx, $O_3$ and VOCs. The oxidation rate of these gaseous pollutants in the liquid-solid heterogeneous system is much faster than that in gaseous phase. Thus, water film on the protein surface may accelerate the oxidation of the absorbed gaseous pollutants in the atmosphere.

**5 Conclusion**

In this study, the composition profiles of combined amino acids in both the surface soils and dominant plants in the Gobi Desert were characterized. Results indicated that the proteins transported with Gobi Desert dust contain large amounts of alanine, glycine and glutamic acid. Therefore, the concentration of these three CAAs significantly increased and ratios of Ala%/Pro%, Gly%/Pro%, and Glu%/Pro% in $PM_{2.5}$ elevated at four cities on dusty days.

Moreover, $\delta^{15}N$ patterns of CAAs of the surface soil and predominant plants in the Gobi Desert were demonstrated, which were further used to interpretate the source variation in proteinaceous matter in $PM_{2.5}$ at four cities from the non-dust to dust period. The mean $\delta^{15}N$ values of glycine and leucine in the Gobi Desert surface soil were significantly lower than those in urban $PM_{2.5}$ during the non-dust period, which can be used to tracing Gobi dust sources. According to the $\delta^{15}N$ inventories of individual CAAs

in potential emission sources, local plants, surface soil and burning sources were the major sources of CAAs in $PM_{2.5}$ at Beijing, Tianjin, Shijiazhuang, and Taiyuan during the non-dust period. During the dust period, more negative $\delta^{15}N$ values of combined glycine and leucine were observed in $PM_{2.5}$ in Northern China, also suggesting inputs of proteinaceous matter from the Gobi Desert.

The contribution of dust sources to the protein in $PM_{2.5}$ in these downwind regions estimated through

nitrogen isotope mass balance for glycine align with those derived from leucine, indicating that isotopic mass balance based on the $\delta^{15}N$ values of glycine and leucine is an effective tool for assessing the contribution of dust sources to proteinaceous material in $PM_{2.5}$. The results showed that protein-N supplied by Gobi dust can reach 0.36 mg N $m^{-2}$ $d^{-1}$. This large quantity of protein-N deposited in a short period of the year and has important effects on the productivity of oligotrophic ecosystems under the

influence of the East Asian dust belt.

**Competing interests**

The contact author has declared that none of the authors has any competing interests.

**Acknowledgments**

This work was supported by the National Natural Science Foundation of China (Grant Nos. 42363011).

We would like to thank the Global Weather and Climate Information Network (http://www.weatherandclimate.info/) for providing meteorological parameters, including temperature (T), relative humidity (RH) and precipitation, during the sampling period. We would also like to thank the China Air Quality Online Monitoring and Analysis Platform for providing air quality data (https://www.aqistudy.cn/) and NASA's EOSDIS Worldview for providing MODIS satellite images

(https://worldview.earthdata.nasa.gov/).

**Data Availability Statement**

Meteorological parameters, including temperature (T), relative humidity (RH), and windspeed (WD), during the sampling period are available at the following website: http://www.weatherandclimate.info/. Air quality data, including $PM_{2.5}$ and $PM_{10}$ concentrations, are available at the following website: https://www.aqistudy.cn/.

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
