# Peer review of "Asian dust transport proteinaceous matter from the Gobi Desert to Northern China"

_EGUsphere, 2024_

## Referee Comment (RC2)

The paper by Zhu et al. summarizes observations of several amino acids and $\delta^{15}$N of these in PM2.5 in 4 cities in Northern China during a week in March 2018. This period included some days under the influence of dust storms from the Gobe Desert. Along with $\delta^{15}$N of soil and plant samples from the Gobe desert and values of $\delta^{15}$N of amino acids in PM2.5 during non-dust events, they estimated the contribution of Gobe desert dust and plants to proteinaceous material in PM2.5 in these cities during the dusty days. They also estimated deposition rate of amino acids in these regions and concluded that the Gobe desert can be a significant source of proteinaceous N for downwind regions, potentially influencing biogeochemical cycling of N and delivery of nutrients.

The results are interesting, but a discussion about uncertainties and limitations of the results are needed. The paper is overall well-written. I support its publication after the following concerns are addressed:

The main limitation of the work in my opinion is about sample representativeness. Line 110: how many soil and plant samples from Gobe were analyzed? Was there any difference in the results from soil samples at different depths within the 0-10 cm? What area of the desert were the samples collected from? Only one latitude and longitude in indicated in the text. How representative are the samples? Similarly, how many local samp. A table summarizing, number of different samples, average and standard deviation of the values determined in these samples, and a map showing location of the collected samples in Gobe and each city are needed (in SI). How confident can we be because of this limitation in the estimated fraction of proteinaceous PM2.5 originating from Gobe Desert?

1. Line 1 of abstract: Particulate matter transported in dust storms can influence biogeochemical cycles of many elements and not just nitrogen so I suggest removing the reference to nitrogen in this introductory sentence.
2. Line 40, I'm not sure how presence of primary particles from proteinaceous material can affect new particle formation. Can you please clarify?
3. Line 75, define GLY
4. Line 84: "…representative urban centers…"
5. Figure S1: The figure lacks geographical references (i.e., borders, city markers with legends, etc) to guide the reader to the relative location of dust sources in Gobe and receptor sites. Also, please add the color scale.
6. Line 106: remove "1 from"
7. Section 2.2: details on extraction efficiency of the developed methods need to be discussed
8. Line 159: This sentence is not clear to me. Based on the previous sentence, I thought concentration of asparagine and glutamine cannot be determined, but total concentration of asparagine+ aspartic acid and glutamine+ glutamic acid can be. Is that not the case?
9. Line 169: As you mention, deposition velocity for particles is size dependent. What ranges of Vd is expected for the larger sizes of fine aerosols that are the focus of this paper? How much uncertainty would this bring to the estimates of deposition fluxes calculated for the different cities?
10. Figure 1. Are the indicate date stamps indicating midnight or noon? Please clarify in the caption. Either way, it doesn't look like the peak in PM10 in Shijiazhuang occurred 11:00 to 18:00.
11. Line 239: I believe the reference here should be to Figure 2
12. Line 261: consider changing 'increment' to 'increase'

13. Figure 4: Please indicate in the caption what the gray border indicates. I'm also confused if the PM2.5 CAA data are from non-dusty days or from all days?! The caption indicates that CAA to the left of the blue dashed line are depleted in 15N compared to Gobe soil, but that's not the case at all sites (e.g., Ala and Val values are very similar to Gobe for TJ and TY)

14. Figure 5. The lower case alphabets supposedly indicate statistical significance by ANOVA, but what is the difference between a, b, and ab? Please explain further in the caption.

15. Line 339-342: The data in Figure 4 do not support the conclusions mentioned in these sentences: "During the non-dust period, δ15N-CAAs patterns in PM2.5 at four urban sites were generally consistent, with glycine, leucine, isoleucine, alanine and valine exhibiting relatively higher δ15N values than other CAA species (Figure 4, left side)." This is not the case at all the sites. "Besides that, δ15N values of individual CAAs in PM2.5 all fell within their respective ranges observed in local dominant plants, soil, and burning sources at four cities (Figure 4, left side)." And this is not the case for all the CAAs. Please modify the sentences to be consistent with the data.

16. Line 391: add standard deviations to the numbers to indicate uncertainty

17. Figure 8- add city names to the map. Also add standard deviations or actual ranges for the numbers in the table.

18. Line 407: FAA?

19. Line 415: Consider adding "between the total CAAs in PM2.5"

20. Line 438: I believe the reference is to Figure 3

21. Line 451: "would be more negative". Indicate compared to what?

22. Line 478-482: Add a reference to the previous work that's mentioned here. Also, what variability in changes in $\delta^{15}$N was observed through combustion? Please add standard deviations and carry that variability to the estimated values of the local burning samples.

23. Line 499: what specific amino acid was used to provide these estimates? Is that Gly? What are these two amino acids only used for determining the fraction? Is it because they have the lowest $\delta^{15}$N ? Please explain.

24. Line 538: "were elevated"

---

## Author Comment (AC1)

**Dear Reviewer:**

**Thank you for your letter and comments concerning our manuscript EGUSPHERE-2024-2065 entitled "Dust storms transport proteinaceous matter from the Gobi Desert to Northern China". Those comments are all valuable and very helpful improving our paper. As suggested by Reviewer 1, the terms "dust storm" and "dust event" were used interchangeably throughout the manuscript. To maintain consistency, the revised manuscript exclusively uses the term "Asian dust events." Consequently, the title has been updated to "Asian Dust Transport of Proteinaceous Matter from the Gobi Desert to Northern China." We have incorporated your suggestions and revised the manuscript according to the points as follows.**

**Anonymous Referee #1**

The manuscript addresses an interesting and emerging issue regarding the role of dust storms in aerosol transport, its effect on particulate matter composition, and its role in critical environmental issues such as air quality and ecosystem productivity. Specifically, the authors assessed the contribution of Gobi dust to proteinaceous combined amino acids (CAAs) in PM2.5 across four urban regions in Northern China during dust events. The authors collected data from multiple sites (Beijing, Tianjin, Shijiazhuang, Taiyuan), which allows for a regional comparison of CAAs in PM2.5. Their approach involved analyzing the concentrations and $\delta15N$ isotopic signatures of CAAs from both the Gobi desert and local urban sources. Additionally, they quantified the Gobi dust's input to CAAs in PM2.5 and evaluated the dry deposition fluxes of protein-N to explore the biogeochemical impacts. Overall, the rationale for this study is well stated, the experiment is well described, and the work points out important issues that are highly relevant. However, the manuscript needs some improvement.

**Major Comments**
**Q1: The comparison of protein characteristics in PM2.5 from urban environments with those in dust sources lacks a detailed interpretation of its broader implications, being limited only to surface-level comparison. The study could benefit from a more in-depth discussion of the potential biological or chemical mechanisms by which dust-borne CAAs affect the local and regional nitrogen cycle. This could help link the findings to broader environmental implications.**
Answer: Thank you for your suggestion. A more discussion of the potential biological or chemical mechanisms by which dust-borne CAAs affect the local and regional nitrogen cycle were added in the manuscript. Please refer to Section 4.2, 4.3 and 4.4 in discussion.
Three biological and chemical mechanisms may explain the variation in protein characteristics in $PM_{2.5}$ during the dust period, as discussed in the revised manuscript.
First, the correlation of total CAAs concentrations in $PM_{2.5}$ with the local meteorological conditions

(including temperature, humidity or wind speed) and $PM_{10}$ concentration (Table S4) showed that the increase concentration of CAAs in PM2.5 during the dust period was not influenced by local meteorological conditions, but were directedly linked to the long-transported Gobi dust source. Then, we used compound-specific nitrogen isotope analyses of individual CAAs to estimate the contribution of protein from dust sources and its dry deposition flux. The results indicated a decreasing trend in the dry deposition of protein-N along the transport pathway. This suggests that the increased airborne protein concentration in downwind areas during dust events may be depend on the dry deposition of dust particles, with protein either coagulating or condensing onto them. Proteinaceous matter in the atmosphere has been shown to serve as a bioavailable nitrogen source for plants and microorganisms (Ho et al., 2015; Samy et al., 2013; Zhang and Anastasio, 2003). Thus, protein-N from the Gobi Desert may act as a nutrient source for marine ecosystems, significantly contributing to ecological processes in China's marginal seas, such as promoting phytoplankton fertilization and stimulating nitrogen fixation (Duarte et al., 2006; Ho et al., 2015). Our results also indicate that the protein-N inputs from the Gobi Desert are comparable to, or even exceed, the dry deposition of WSON in these regions. The recurring dust events and the increased Asian-supplied protein-N during spring, coinciding with the algae bloom season, may have widespread effects on the productivity of downwind ecosystems, influenced by the East Asian dust belt.

Second, during long-range transport, hydrophilic protein particles mix with mineral particles to form various internal mixtures. These mixed particles can absorb water efficiently, thereby increasing CCN activity and altering optical properties, which is important for understanding global climate and hydrological cycles (Adachi et al., 2020).

Third, aerosol proteins increase might affect the generation of secondary aerosols (Li et al., 2020). Under high relative humidity condition, water might condense onto protein aerosols due to their good hygroscopicity, resulting in forming a film of water on the aerosol surface. This water film is easy to adsorb gaseous pollutants in the atmosphere, including $SO_2$, NOx, $O_3$ and VOCs. The oxidation rate of these gaseous pollutants in the liquid-solid heterogeneous system is much faster than that in gaseous phase. Thus, water film on the protein surface may accelerate the oxidation of the absorbed gaseous pollutants in the atmosphere.

Thank you very much for your suggestions. These revisions have provided us with a more accurate understanding of the factors and mechanisms influencing protein concentrations in the atmospheres of downwind cities during dust transport, which has enhanced the environmental implications of this study.

**Q2: The sampling period (March 24–31, 2018) represents a short window of time. While the authors acknowledge the occurrence of dust storms during this period, it is unclear if these results are representative of long-term trends or typical dust events. A broader period could strengthen the conclusions, or could the authors provide additional context on whether this period represents typical dust activity for that year or region.**
Answer: Thank you for your suggestion. Globally, the primary sources of mineral dust are the arid regions of North Africa, the Arabian Peninsula, Central Asia, and Northeast Asia (Filonchyk, 2022;

Zhou et al., 2019). East Asia is the second-largest dust source region worldwide, with annual dust emissions estimated at 214 Tg yr$^{-1}$ (Tian et al., 2020). Furthermore, the westerlies in the Northern Hemisphere's middle latitudes can transport Asian dust from upwind areas to distant downwind regions, potentially completing a global cycle and exerting far-reaching impacts (Xie et al., 2023; Zhou et al., 2019). Asian dust particles primarily originate from the arid and semi-arid areas of northwestern China and Mongolia, including the Taklamakan Desert, the Gobi Desert, the Badan Jaran Desert, the Tengger Desert, and others (Shao and Dong, 2006). Recent research indicates that the Gobi Desert, rather than the Taklamakan Desert, is the primary contributor to dust concentrations in East Asia in spring (Chen et al., 2017; Tang et al., 2018). Spring is considered the peak season for sand and dust storms in Northeast Asia, as positive surface pressure anomalies over the Tamil Peninsula intensify cold air outbreaks across the desert regions of northwestern China and Mongolia (Yang et al., 2008). A particularly intense and widespread dust event occurred between March 26–29, 2018, in the North China Plain, which was the most significant dust storm in recent years in China (Zhou et al., 2019). This dust event affected nearly two-thirds of China and parts of the Northwest Pacific (Tian et al., 2020). Therefore, PM$_{2.5}$ sampling at four representative sites (Beijing, Tianjin, Shijiazhuang, and Taiyuan) located in the downwind areas of the Gobi Desert during the 26–29 March 2018 dust event provides a typical representation of Asin dust activity in northern China.

The content has been incorporated into the introduction section of the revised manuscript. Line 83~99.

**Q3: The use of satellite imagery and back-trajectory analysis to confirm Gobi dust as the source is an important strength of the paper. However, additional validation through ground-based measurements or comparison with other dust episodes would confirm that the identified CAA increases were directly attributable to Gobi dust.**

Answer: Thank you for your suggestion. Sorry for our unclear description. In this study, the conclusion that the identified CAA concentrations in the North China Plain increase during the dust period and are directly attributable to Gobi dust is based on ground-based observations. Satellite imagery and back-trajectory are auxiliary tools in this study. Surface soil, vegetation, and PM2.5 samples were collected from both the North China Plain and the Gobi Desert source area. The concentrations and δ$^{15}$N values of individual CAAs in these samples were measured. By comparing the concentrations, percentage compositions, and nitrogen isotopes of individual CAAs between dust and non-dust periods, we found that CAAs transported by Gobi dust were rich in alanine, glycine, and glutamic acid. The concentrations and percentages of these three CAAs in PM2.5 from Northern China notably increased during dust periods. From the non-dust to dust periods, glycine and leucine in urban PM2.5 exhibited negative shifts in their δ$^{15}$N values, confirming that Gobi dust is a significant source of CAAs in PM2.5 in Northern China. Thus, this study represents a field-based investigation. We analyzed the variation in the percentage composition and nitrogen isotopes of individual CAAs in PM2.5 samples from the non-dust to dust periods, which confirmed the contribution of Gobi dust to the increased CAA concentrations in downwind areas.

Additionally, we reviewed literature on ground-based dust observations in the North China Plain during our sampling period. Tian et al. (2020) also confirmed that an intensive dust event occurred in the North China Plain from 26–29 March 2018 originating from western Inner Mongolia. Line 281~282.

**Q4: The authors use a fixed dry deposition velocity (Vd) based on previous studies. However, Vd is typically influenced by factors such as particle size, wind speed, and hygroscopicity. The use of a fixed value introduces uncertainty in the estimation of deposition fluxes. A site-specific Vd estimate would improve the accuracy of their calculations.**

Answer: Thank you for your suggestion. Indeed, the use of a fixed value introduces uncertainty in the estimation of deposition fluxes. However, since we did not obtain data on the size distribution of CAAs, site-specific Vd cannot be obtained. In future research, we will further study the particle size distribution characteristics of atmospheric proteins to more accurately calculate the atmospheric protein-N deposition flux. In the revised manuscript, we have provided the uncertainty in protein deposition flux caused by the uncertainty of dry deposition velocity of protein-N (Figure 8).

Primary biological aerosols have been found to be distributed from nanometers up to about a tenth of a millimeter, with their size distribution influenced by their sources (Fröhlich-Nowoisky et al., 2016). Therefore, in this study, the deposition velocities of protein-N were assumed to be the same as those used to estimate water-soluble nitrogen dry deposition (0.012 m s$^{-1}$), given that protein-N is a significant component of water-soluble organic nitrogen (WSON) in aerosols and WSON has also been detected in both coarse and fine fractions (Zamora et al., 2011; Zhang and Anastasio, 2003). The uncertainty in the value for the dry deposition velocity can lead to the uncertainty in dry flux estimates. For particles in the size range where gravitational setting is the controlling factor, Vd values obtained by model and field experiment were consistent (Spokes et al., 2000). (Duce et al., 1991) reported that under wind speeds ranging from 0 to13 m s$^{-1}$ and relative humidity between 0% and 100%, the deposition velocity for submicrometer aerosol particles is 0.1 cm s-1 ± a factor of 3, while the deposition velocity for supermicrometer crustal particles is 1 cm s-1, also with an uncertainty factor of 3. During the dust period, the wind speeds in Beijing, Tianjin, Shijiazhuang, and Taiyuan were 2.0~2.4 m s$^{-1}$, 4.1~6.1 m s$^{-1}$, 1.7~2.6 m s$^{-1}$ and 4.6~7.0 m s$^{-1}$, respectively, and the relative humidity were 25.8~28.4%, 24.3~28.0%, 24.3~29.1%, and 37.0~38.5%, respectively. The variations in wind speed and relative humidity across the four sampling cities were relative minor, and their ranges fell within those reported by (Duce et al., 1991).

Based on this, the uncertainty for the deposition velocity of aerosol protein-N in this study was set to a factor of 3. Line 198~216. Using the formula for error propagation for multiplication:

$$\left(\frac{\Delta Fdry}{Fdry}\right)^2 = \left(\frac{\Delta C}{C}\right)^2 + \left(\frac{\Delta Vd}{Vd}\right)^2$$

Since the concentration C is measured with negligible uncertainty (0.1) compared to Vd, the term $\triangle C/C$ can be considered zero. Therefore, the equation simplifies to:

$$\frac{\Delta Fdry}{Fdry} = \frac{\Delta Vd}{Vd}$$

Given $\triangle Vd/Vd = 3$:

$$\frac{\Delta Fdry}{Fdry} = 3$$

Consequently, Fdry will also have the same factor of uncertainty:

$$F_{dry,min} = C \cdot \frac{Vd}{3}$$

$$F_{dry,max} = C \cdot 3 \cdot Vd$$

Each deposition flux value of protein-N with the uncertainty range at the four sampling cities was provided in the revised manuscript (Figure 8).

[Figure]

Figure 8. The dry deposition flux of protein-N at BJ, TJ, SJZ and TY during the dust period (blue bar) and published reports of the dry deposition flux of WSON measured in different atmospheric scenarios (red bar). The length of the bars represents the dry deposition flux of WSON and protein-N. The WSON deposition data were sourced from previous studies (Shi et al., 2010; Ho et al., 2015; Matsumoto et al., 2014; Tsagkaraki et al., 2021; Nakamura et al., 2006; Zhang et al., 2011). The method of calculating deposition fluxes of protein-N with the uncertainty range at the four sampling sites was provided in the supplementary manuscript (Text 1).

**Q5: The paper focuses heavily on the Gobi dust contribution but does not extensively explore the potential influence of local urban sources of CAAs. It would be helpful to discuss how industrial, vehicular, or other anthropogenic emissions contribute to the CAAs in PM$_{2.5}$. This would provide a clearer differentiation between dust and urban source contributions. Further justification or consideration of urban δ15N variability is needed.**

Answer: Thank you for your suggestion. We have added more discussion focused on the local urban sources contributing to the CAAs in aerosols in the revised manuscript. A new section, "4.1 Local Urban Sources of CAAs in PM$_{2.5}$," has been added to discuss these sources. The concentration of CAAs measured during the non-dust period was compared with previous studies. The δ$^{15}$N values of individual CAAs in PM$_{2.5}$ from four cities were compared to their respective values in local common plants, soil, and burning sources across the four urban sites to further identify the local

sources of CAAs in $PM_{2.5}$. Additionally, we compared our results of protein source apportionment with previous reports, finding no discrepancies. The results confirmed that local dominant plants, surface road dust, and anthropogenic industrial activities involving combustion processes were the major sources of CAAs in $PM_{2.5}$ across Beijing, Tianjin, Shijiazhuang, and Taiyuan during the non-dust period.

Please refer to Line 454~484.

**Minor Comments**

**Q6: It is suggested that the sampling locations be visually represented on a map and be incorporated into Fig 8 as a subplot. This will provide a clearer spatial understanding of the study's geographic scope.**

Answer: Thank you for your suggestion. The reviewer 2 also made a similar suggestion. A map showing the locations of the collected samples in the Gobi Desert and each city has been added in Supplementary Material (Figure S1).

**Q7: The terms dust storm and dust event are used interchangeably throughout the manuscript, but they can be interpreted as two different transport mechanisms in the dust transport field. The author should consistently use one or the**

Answer: Thank you for your suggestion. The term "dust storm" has been revised to "Asian dust event" throughout the revised manuscript.

Reviewer 2

**Q8: The main limitation of the work in my opinion is about sample representativeness. Line 110: how many soil and plant samples from Gobe were analyzed? Was there any difference in the results from soil samples at different depths within the 0-10 cm? What area of the desert were the samples collected from? Only one latitude and longitude in indicated in the text.How representative are the samples? Similarly, how many local samp. A table summarizing, number of different samples, average and standard deviation of the values determined in these samples, and a map showing location of the collected samples in Gobe and each city are needed (in SI). How confident can we be because of this limitation in the estimated fraction of proteinaceous PM2.5 originating from Gobe Desert?**

Answer: Thank you for your suggestion. A table summarizing, number of different samples, average and standard deviation of the values determined in these samples (Table S1) and a map showing location of the collected samples in Gobi and each city (Figure S1) have been included in the supplementary materials. Surface soil samples from the Gobi Desert were collected from an area ranging between 43.46°N to 43.60°N and 112.00°E to 112.05°E, covering approximately 45 km², as shown in Figure S1. The Gobi Desert is the major source of sand for dust storms in Asia during the spring (An et al., 2013). Therefore, five sampling sites along the transport pathway of the dust event that occurred from March 26 to 29, 2018 (Figure 2), were selected to represent the Erenhot Gobi Desert. Each site was free of anthropogenic interference. Surface sand samples, which are most likely to be aerosolized, were collected from the tops of dunes using a plastic spatula and

stored in sealed plastic bags until transported to the laboratory. At each location, surface soil was collected from five randomly selected sampling points within a radius of approximately 20 cm. These five sub-samples were then combined to create one representative sample. Line 129~137. Apologies for the confusion. By "0–10 cm," we refer to the surface soil samples that were collected. During dust events, the surface sand samples from the tops of dunes are the most likely to be aerosolized. Therefore, at all sampling locations, only surface soil was collected, following the method outlined by (An et al., 2013). Therefore, "0–10 cm," was changed to "the surface soil" in revised manuscript.

Five surface soil and twelve dominate plant samples were collected from the Gobi Desert (Table S1).

**Q9: Line 1 of abstract: Particulate matter transported in dust storms can influence biogeochemical cycles of many elements and not just nitrogen so I suggest removing the reference to nitrogen in this introductory sentence.**

Answer: Thank you for your suggestion. "nitrogen" was deleted in this sentence.

**Q10: Line 40, I'm not sure how presence of primary particles from proteinaceous material can affect new particle formation. Can you please clarify?**

Answer: Thank you for your suggestion. In a previous study, (Li et al., 2022) suggested that aerosol proteins might affect the generation of secondary aerosols. They proposed that under haze condition, the relative humidity increases and then high humidity facilitates water condensation onto protein aerosols due to their good hygroscopicity, resulting in forming a film of water on the aerosol surface. This water film is easy to adsorb gaseous pollutants in the atmosphere, including $SO_2$, $NO_X$, $NH_3$, $O_3$ and VOCs. It is well known that the oxidation rate of gaseous pollutants in the liquid-solid heterogeneous system is much faster than that of gaseous phase. The reaction of $O_3$ in wet particles generates strong oxidizing substances such as hydroxyl free radical ($\cdot OH$), which accelerated the oxidation of the absorbed $SO_2$ and $NO_X$ into $SO_4^{2-}$ and $NO_3^-$. During these processes, some parts of the protein core may be oxidized or nitrified protein through nitrification reaction (Liu et al., 2017; Shiraiwa et al., 2012).

**Q11: Line 75, define GLY**

Answer: Thank you for your suggestion. GLY (glycine) was defined in the revised manuscript.

**Q12: Line 84: "⋯representative urban centers⋯"**

Answer: Thank you for your suggestion. It has been revised as your suggestion.

**Q13: Figure S1: The figure lacks geographical references (i.e., borders, city markers with legends, etc) to guide the reader to the relative location of dust sources in Gobe and receptor sites. Also, please add the color scale.**

Answer: Thank you for your suggestion. Geographical references including borders, city markers with legends and the color scale have been added.

**Q14: Line 106: remove "1 from"**

Answer: Sorry for our mistake. "1 from" was deleted.

**Q15: Section 2.2: details on extraction efficiency of the developed methods need to be discussed**
**Answer: Thank you for your suggestion. The extraction efficiency of the method was added in the supplement material.**

Answer: Thank you for your suggestion. To evaluate the extraction efficiency, analytical method was applied to the samples were spiked with the amino acid standard mixtures ($100\mu l$ $1nmol$ $\mu l^{-1}$). The average recovery ratios of the individual amino acids were shown in Table S2. The recoveries for the majority of CAAs ranged from 80.7% (tyrosine) to 106.5% (glycine). The precisions of the investigated AAs were better than 10%. Line 170~173.

Table S2. The recoveries and precision of CAA analysis.

| Amino acids | Recovery (%) | Precision (%) |
|---|---|---|
| Glycine (Gly) | 106.5 | 1.1 |
| Alanine (Ala) | 92.6 | 3.6 |
| Aspartic acid (Asp) | 95.0 | 8.6 |
| Glutamic acid (Glu) | 105.6 | 8.8 |
| γ-amino butyric acid (Gaba) | 95.7 | 7.7 |
| Serine (Ser) | 92.7 | 5.7 |
| Proline (Pro) | 93.2 | 6.7 |
| Threonine (Thr) | 90.0 | 6.1 |
| Valine (Val) | 95.5 | 1.3 |
| Lysine (Lys) | 82.5 | 6 |
| Leucine (Leu) | 92.0 | 0.8 |
| Isoleucine (Ile) | 93.1 | 2 |
| Arginine (Arg) | 85.4 | 9.4 |
| Phenylalanine (Phe) | 93.7 | 5.4 |
| Ornithine (Orn) | 91.5 | 9.9 |
| Tyrosine (Tyr) | 80.7 | 0.3 |
| Histidine (His) | 95.1 | 1.7 |
| Methionine (Met) | 86.6 | 2.7 |

**Q16: Line 159: This sentence is not clear to me. Based on the previous sentence, I thought concentration of asparagine and glutamine cannot be determined, but total concentration of asparagine+ aspartic acid and glutamine+ glutamic acid can be. Is that not the case?**

Answer: Sorry for our mistake. The concentration of aspartic acid represents total concentration of asparagine+ aspartic acid and the concentration of glutamic acid represents glutamine+ glutamic acid. It was changed to "Since asparagine and glutamine are converted to aspartic acid and glutamic acid in the hydrolysis process, respectively, the concentration and $\delta^{15}N$ value of combined aspartic acid represents the sum of aspartic acid and asparagine. The concentration and $\delta^{15}N$ value of combined glutamic acid represents the sum of glutamic acid and glutamine." Line 186-187.

**Q17: Line 169: As you mention, deposition velocity for particles is size dependent. What ranges of Vd is expected for the larger sizes of fine aerosols that are the focus of this paper? How much uncertainty would this bring to the estimates of deposition fluxes calculated for the different cities?**

Answer: Thank you for your suggestion. The uncertainty in the value for the dry deposition velocity can lead to the uncertainty in dry flux estimates. For particles in the size range where gravitational setting is the controlling factor, Vd values obtained by model and field experiment were consistent (Spokes et al., 2000). (Duce et al., 1991) reported that under wind speeds ranging from 0 to13 m s$^{-1}$ and relative humidity between 0% and 100%, the deposition velocity for submicrometer aerosol particles is 0.1 cm s-1 ± a factor of 3, while the deposition velocity for supermicrometer crustal particles is 1 cm s-1, also with an uncertainty factor of 3. During the dust period, the wind speeds in Beijing, Tianjin, Shijiazhuang, and Taiyuan were 2.0~2.4 m s$^{-1}$, 4.1~6.1 m s$^{-1}$, 1.7~2.6 m s$^{-1}$ and 4.6~7.0 m s$^{-1}$, respectively, and the relative humidity were 25.8~28.4%, 24.3~28.0%, 24.3~29.1%, and 37.0~38.5%, respectively. The variations in wind speed and relative humidity across the four sampling cities were relative minor, and their ranges fell within those reported by (Duce et al., 1991). **Based on this, the uncertainty for the deposition velocity of aerosol protein-N in this study was set to a factor of 3. Therefore, Vd ranged from 0.004 m/s to 0.036 m/s.** Since Fdry=C·Vd, the formula was used for error propagation for multiplication:

$$\left(\frac{\Delta Fdry}{Fdry}\right)^2 = \left(\frac{\Delta C}{C}\right)^2 + \left(\frac{\Delta Vd}{Vd}\right)^2$$

Since the concentration C is measured with negligible uncertainty (0.1) compared to Vd, the term △C/C can be considered zero. Therefore, the equation simplifies to:

$$\frac{\Delta Fdry}{Fdry} = \frac{\Delta Vd}{Vd}$$

Given △Vd/Vd = 3:

$$\frac{\Delta Fdry}{Fdry} = 3$$

Consequently, Fdry will also have the same factor of uncertainty:

$$F_{dry,min} = C \cdot \frac{Vd}{3}$$

$$F_{dry,max} = C \cdot 3 \cdot Vd$$

Each deposition flux value of protein-N with the uncertainty range at the four sampling cities was provided in the revised manuscript (Figure 8). Line 198~216.

**Q18: Figure 1. Are the indicate date stamps indicating midnight or noon? Please clarify in the caption. Either way, it doesn't look like the peak in PM10 in Shijiazhuang occurred 11:00 to 18:00.**

Answer: Thank you for your suggestion. The timestamps indicate 21:00. This was added in the caption of figure 1. With the consideration of the timestamps is 21:00, the peak in PM10 in Shijiazhuang occurred 11:00 to 18:00.

**Q19: Line 239: I believe the reference here should be to Figure 2**

Answer: Sorry for our mistake. It has been revised as your suggestion.

**Q20: Line 261: consider changing 'increment' to 'increase'**
Answer: Thank you for your suggestion. It was revised to "increase".

All changes can be tracked in the revised manuscript. Thank you very much again.

Yours sincerely,

Ren-Guo Zhu, Hua-Yun Xiao, Meiju Yin, Hao Xiao, Zhongkui Zhou, Yuanyuan Pan, Guo Wei, Cheng Liu

**Reference**
Adachi, K., Oshima, N., Gong, Z., De Sá, S., Bateman, A. P., Martin, S. T., De Brito, J. F., Artaxo, P., Cirino, G. G., Sedlacek Iii, A. J., and Buseck, P. R.: Mixing states of Amazon basin aerosol particles transported over long distances using transmission electron microscopy, Atmospheric Chem. Phys., 20, 11923–11939, https://doi.org/10.5194/acp-20-11923-2020, 2020.

An, S., Couteau, C., Luo, F., Neveu, J., and DuBow, M. S.: Bacterial Diversity of Surface Sand Samples from the Gobi and Taklamaken Deserts, Microb. Ecol., 66, 850–860, https://doi.org/10.1007/s00248-013-0276-2, 2013.

Chen, S., Huang, J., Li, J., Jia, R., Jiang, N., Kang, L., Ma, X., and Xie, T.: Comparison of dust emissions, transport, and deposition between the Taklimakan Desert and Gobi Desert from 2007 to 2011, Sci. China Earth Sci., 60, 1338–1355, https://doi.org/10.1007/s11430-016-9051-0, 2017.

Duce, R. A., Liss, P. S., Merrill, J. T., Atlas, E. L., Buat-Menard, P., Hicks, B. B., Miller, J. M., Prospero, J. M., Arimoto, R., Church, T. M., Ellis, W., Galloway, J. N., Hansen, L., Jickells, T. D., Knap, A. H., Reinhardt, K. H., Schneider, B., Soudine, A., Tokos, J. J., Tsunogai, S., Wollast, R., and Zhou, M.: The atmospheric input of trace species to the world ocean, Glob. Biogeochem. Cycles, 5, 193–259, https://doi.org/10.1029/91GB01778, 1991.

Filonchyk, M.: Characteristics of the severe March 2021 Gobi Desert dust storm and its impact on air pollution in China, Chemosphere, 287, 132219, https://doi.org/10.1016/j.chemosphere.2021.132219, 2022.

Fröhlich-Nowoisky, J., Kampf, C. J., Weber, B., Huffman, J. A., Pöhlker, C., Andreae, M. O., Lang-Yona, N., Burrows, S. M., Gunthe, S. S., Elbert, W., Su, H., Hoor, P., Thines, E., Hoffmann, T., Després, V. R., and Pöschl, U.: Bioaerosols in the Earth system: Climate, health, and ecosystem interactions, Atmospheric Res., 182, 346–376, https://doi.org/10.1016/j.atmosres.2016.07.018, 2016.

Li, Y., Haoyue, Z., Aotang, L., Jiali, Z., and Shengli, D.: High time-resolved variations of proteins in PM2.5 during haze pollution periods in Xi'an, China, Environ. Pollut., 305, 119212, https://doi.org/10.1016/j.envpol.2022.119212, 2022.

Liu, F., Lai, S., Tong, H., Lakey, P. S. J., Shiraiwa, M., Weller, M. G., Pöschl, U., and Kampf, C. J.: Release of free amino acids upon oxidation of peptides and proteins by hydroxyl radicals, Anal. Bioanal. Chem., 409, 2411–2420, https://doi.org/10.1007/s00216-017-0188-y, 2017.

Shao, Y. and Dong, C. H.: A review on East Asian dust storm climate, modelling and monitoring, Glob. Planet. Change, 52, 1–22, https://doi.org/10.1016/j.gloplacha.2006.02.011, 2006.

Shiraiwa, M., Selzle, K., Yang, H., Sosedova, Y., Ammann, M., and Pöschl, U.: Multiphase chemical

kinetics of the nitration of aerosolized protein by ozone and nitrogen dioxide, Environ. Sci. Technol., 46, 6672–6680, https://doi.org/10.1021/es300871b, 2012.

Spokes, L. J., Yeatman, S. G., Cornell, S. E., and Jickells, T. D.: Nitrogen deposition to the eastern Atlantic Ocean. The importance of south-easterly flow, Tellus B, 52, 37–49, https://doi.org/10.1034/j.1600-0889.2000.00062.x, 2000.

Tang, K., Huang, Z., Huang, J., Maki, T., Zhang, S., Shimizu, A., Ma, X., Shi, J., Bi, J., Zhou, T., Wang, G., and Zhang, L.: Characterization of atmospheric bioaerosols along the transport pathway of Asian dust during the Dust-Bioaerosol 2016 Campaign, Atmospheric Chem. Phys., 18, 7131–7148, https://doi.org/10.5194/acp-18-7131-2018, 2018.

Tian, Y., Pan, X., Wang, Z., Wang, D., Ge, B., Liu, X., Zhang, Y., Liu, H., Lei, S., Yang, T., Fu, P., Sun, Y., and Wang, Z.: Transport Patterns, Size Distributions, and Depolarization Characteristics of Dust Particles in East Asia in Spring 2018, J. Geophys. Res. Atmospheres, 125, e2019JD031752, https://doi.org/10.1029/2019JD031752, 2020.

Xie, W., Fan, C., Qi, J., Li, H., Dong, L., Hu, W., Kojima, T., and Zhang, D.: Decrease of bioaerosols in westerlies from Chinese coast to the northwestern Pacific: Case data comparisons, Sci. Total Environ., 864, 161040, https://doi.org/10.1016/j.scitotenv.2022.161040, 2023.

Yang, Y. Q., Hou, Q., Zhou, C. H., Liu, H. L., Wang, Y. Q., and Niu, T.: Sand/dust storm processes in Northeast Asia and associated large-scale circulations, Atmospheric Chem. Phys., 8, 25–33, https://doi.org/10.5194/acp-8-25-2008, 2008.

Zamora, L. M., Prospero, J. M., and Hansell, D. A.: Organic nitrogen in aerosols and precipitation at Barbados and Miami: Implications regarding sources, transport and deposition to the western subtropical North Atlantic, J. Geophys. Res., 116, D20309, https://doi.org/10.1029/2011JD015660, 2011.

Zhang, Q. and Anastasio, C.: Free and combined amino compounds in atmospheric fine particles (PM2.5) and fog waters from Northern California, Atmos. Environ., 37, 2247–2258, https://doi.org/10.1016/S1352-2310(03)00127-4, 2003.

Zhou, C., Gui, H., Hu, J., Ke, H., Wang, Y., and Zhang, X.: Detection of New Dust Sources in Central/East Asia and Their Impact on Simulations of a Severe Sand and Dust Storm, J. Geophys. Res. Atmospheres, 124, 10232–10247, https://doi.org/10.1029/2019JD030753, 2019.

---

## Author Response (AR2)

Dear editor Roya,

Thank you for your email and for giving us another opportunity to revise our manuscript. We sincerely appreciate your constructive comments. In response to your concerns regarding potential changes in background isotopic values and their associated uncertainties, we have carefully addressed these issues in our revision. Specifically, we have incorporated a multi-isotope MixSIAR model for comparative analysis in Taiyuan. Below is our detailed response to your comments:

**Major Comments**
Please add a discussion to the manuscript on the limitations of the sample size, statistical significance tests for the results tabulated in Table S1, and a discussion of uncertainties/caveats/biases that the results have, especially during the dust evens and at sites where the local isotopic signatures have a similar mean to the various Gobe desert source samples.

Answer: Thank you for your suggestion. A more discussion of uncertainties/caveats/biases that the results have been added in revised manuscript. Please refer to Section 4.3. Figures S3, S4 and S5 as well as stable isotope analysis using MixSIAR in R (Text 2) have been added in supplementary Materials.

In this study, we applied a two-endmember mixing model comparing compound-specific nitrogen isotopic values of CAAs from Gobi dust source with those in urban aerosols during non-dust periods (representing the atmospheric background values), rather than those in local dominant plants, road dust and anthropogenic activities sources. Our methodology relies on the fundamental assumption that the nitrogen isotopic composition of CAAs in urban aerosols during the dust period, which derived from local dominant plants, road dust and anthropogenic activities, does not significantly differ from background atmospheric values.

For the Beijing, Tianjin and Shijiazhuang sampling sites, meteorological conditions (wind speed, relative humidity, and temperature) did not differ significantly ($p > 0.05$) between dust and non-dust periods (Figures S3 and S4). Under these stable conditions, local urban emission sources remained consistent, maintaining unchanged atmospheric background values. Therefore, the application of a two-endmember mixing model - utilizing isotopic values of compound-specific CAAs from Gobi dust sources and urban aerosols during non-dust periods - provides a scientifically robust approach for quantifying Gobi dust contributions in Beijing, Tianjin and Shijiazhuang.

At the Taiyuan sampling site, wind speeds during dust events were significantly higher than during non-dust periods ($p < 0.05$; Figures S3–S4). These strong winds may enhance entrainment of local plant debris and road dust into aerosols, potentially modifying baseline $\delta^{15}N$ signatures of CAAs. This suggests CAA $\delta^{15}N$ values during the non-dust period (atmospheric background) may not fully represent local signatures during dust events, potentially affecting source apportionment. To evaluate this effect, we applied the MIXSIAR model (Stock and Semmens, 2016; Song et al., 2021) with $\delta^{15}N$ values of both glycine and leucine from local and the Gobi dust sources (Table S1, details were provided in Supplementary Materials, Text2). The MIXSIAR model showed that in Taiyuan, the relative contributions of local dominant plants, road dust, and anthropogenic activities sources to aerosol CAAs averaged $1.6 \pm 2.6\%$, $46.2 \pm 20.9\%$ and $45.8 \pm 12.2\%$, respectively, during the non-dust period. The dust period exhibited modified contribution profiles: local plants

(12.6±10.3%), road dust (32.7±19.8%), and anthropogenic emissions (35.9±10.0%), along with an external contribution from Gobi dust sources (18.7±14.1%). The Gobi dust contribution estimated by MIXSIAR showed agreement with our two-endmember model estimates (17 ± 25% ~ 23 ± 10%). After normalizing MIXSIAR results to exclude Gobi dust contributions, natural sources (local plant + road dust) showed only a modest increase from 47.8% (non-dust) to 55.8% (dust periods), representing an 8% enhancement.

During this dust sampling campaign, only the Taiyuan site exhibited significant wind speed increases, while other sampling locations showed no remarkable meteorological variations. Future research should incorporate more extended observational periods of dust events, with particular emphasis on downwind areas experiencing significant meteorological changes. Such extended investigations will enable more accurate assessment of how long-range transported dust sources influence biogeochemical cycles in downwind ecosystems.

Besides that, uncertainties associated in the two-endmember mixing model analysis, which derived from the $\delta^{15}N$ variabilities of combined Gly and Leu in emission sources (mean ± SD $\delta^{15}N$ values) were added in our revised manuscript. Please refer to line 607~608. The contributions of dust sources to Beijing, Tianjin, Shijiazhuang, and Taiyuan during this dust event were calculated for glycine as 94 ± 17%, 78 ± 7%, 36 ± 1%, and 17 ± 25% respectively. For leucine, the contributions were 98 ± 23%, 83 ± 11%, 44 ± 12%, and 23 ± 10%, respectively.

Stable Isotope Analysis in R
The Bayesian mixing models in R model (MixSIAR) has been widely used to determine diet composition, population structure, and animal movement, because it can use the isotope values (biotracer data) to estimate the proportions of source contributions to a mixture, and incorporate the uncertainties associated with source isotope values (Stock, and Semmens, 2016, Song et al., 2021). In our study, the MixSIAR model with $\delta^{15}N$ values of both glycine and leucine was used to incorporate source apportionment of CAAs in Taiyuan during the non-dust period and dust period, respectively. If quantifying dust source contributions using nitrogen isotopic values from individual amino acids (e.g., glycine or leucine alone) by mixing local and Gobi dust sources would introduce larger uncertainty. This limitation stems from the fact that $\delta^{15}N$ values of glycine or leucine in Gobi dust show no statistically significant differences compared to those of urban road soils across all study sites ($p > 0.05$; Figure S5). Therefore, both $\delta^{15}N$ values of both glycine and leucine from local and the Gobi dust sources were used to minimal uncertainty. Details on the assumed values for each potential end-member are of great importance. As discussed in section 4.1, local dominant plants, surface road dust and anthropogenic industrial activities were the major sources of CAAs in $PM_{2.5}$ during the non-dust period. Therefore, three major sources, including local dominant plants, surface road dust and anthropogenic sources were used to simulate the relative contributions of major urban protein sources during non-dust periods in Taiyuan (Table S1). Four distinct source categories were used to calculate their respective contribution to urban proteinaceous matter in $PM_{2.5}$: dust from Gobi Desert, local dominant plants, road dust and anthropogenic activitiy sources in Taiyuan. In our estimations, uncertainties associated in the source analysis were derived from the $\delta^{15}N$ variabilities of major combined Gly and Leu sources (mean ± SD $\delta^{15}N$ values of each source were input; Table S1). The result of MixSIAR model was exhibited in Table S5.

All changes can be tracked in the revised manuscript. Thank you very much again.

Yours sincerely,

Ren-Guo Zhu, Hua-Yun Xiao, Meiju Yin, Hao Xiao, Zhongkui Zhou, Yuanyuan Pan, Guo Wei, Cheng Liu

**References**

Stock, B. C. and B. X. Semmens. MixSIAR GUI User Manual. Version 3.1. https://github.com/brianstock/MixSIAR/. doi:10.5281/zenodo.47719. 2016.

Song, W., Liu, X.-Y., Hu, C.-C., Chen, G.-Y., Liu, X.-J., Walters, W. W., Michalski, G., and Liu, C.-Q.: Important contributions of non-fossil fuel nitrogen oxides emissions, Nat. Commun., 12, 243, https://doi.org/10.1038/s41467-020-20356-0, 2021.

---

## Author Response (AR4)

Response to Editor's Minor Revisions
Manuscript ID: egusphere-2024-2065
Title: Asian dust transport proteinaceous matter from the Gobi Desert to Northern China

Dear Editor Roya,

Thank you for your careful review of our manuscript and for identifying the minor issues in the supplementary materials. We sincerely appreciate your time and the thoughtful suggestions, which have further improved the clarity and precision of our paper. Below is a point-by-point response to your comments:

Comment: there's a sentence in Text 2 that needs to be rephrased: "If quantifying dust source contributions using nitrogen isotopic values from individual amino acids (e.g., glycine or leucine alone) by mixing local and Gobi dust sources would introduce larger uncertainty. "
Response: As suggested, we have rephrased this sentence. "Quantifying dust source contributions using nitrogen isotopic values from single amino acids (e.g., exclusively glycine or leucine) may introduce substantial uncertainty. "

Comment: There's a pink shaded area in the time series. Please add to caption what this highlighted area means (i.e., dust even period)
Response: Thank you for your suggestion. The meaning of the highlighted area has been added to the caption of Figure S4. "The yellow shaded area represents the dust period."

Comment: Fig S5: It's still not clear what the letter by each box and whisker plot mean. Please clarify this in the caption.
Response: Thank you for your suggestion. In the initial version, we employed a two-way ANOVA. As you rightly pointed out, the letter did not clearly present the results, particularly in illustrating isotopic differences between sampling sites. To address this, the revised version now conducts separate one-way ANOVAs for glycine and leucine $\delta^{15}$N values across sampling locations, with corresponding updates to the figure caption for clarity.

"Figure S5. Box-and-whisker plots show the $\delta^{15}$N values of combined Gly and Leu in the surface dust from the Gobi Desert and road dust from four North China Plain. Each box encompasses the $25^{th}$-$75^{th}$ percentiles. Whiskers and hollow squares inside each box are the SD and mean values, respectively. Dots are replicate measurements at each site. Differences in $\delta^{15}$N values of combined Gly and combined Leu between the Gobi Desert and four North China Plain cities were examined using one-way ANOVA with Tukey's HSD test (p < 0.05). Statistically significant differences in mean Gly or Leu values between the Gobi Desert and individual cities are indicated by different letters (Tukey's HSD test)."

All modifications have been implemented in the revised supplementary materials, and the updated files have been re-uploaded to the submission system. These changes do not affect any data or conclusions in the main manuscript.

We sincerely appreciate your attention to these details. Please let us know if any further adjustments

are required. Thank you very much again.

Yours sincerely,
Ren-Guo Zhu, Hua-Yun Xiao, Meiju Yin, Hao Xiao, Zhongkui Zhou, Yuanyuan Pan, Guo Wei, Cheng Liu